# Unsupervised Anomaly Detection with Rejection

**Lorenzo Perini**
DTAI lab & Leuven.AI,
KU Leuven, Belgium
lorenzo.perini@kuleuven.be

**Jesse Davis,**
DTAI lab & Leuven.AI,
KU Leuven, Belgium
jesse.davis@kuleuven.be

## Abstract

Anomaly detection goal is to detect unexpected behaviours in the data. Because anomaly detection is usually an unsupervised task, traditional anomaly detectors learn a decision boundary by employing heuristics based on intuitions, which are hard to verify in practice. This introduces some uncertainty, especially close to the decision boundary, which may reduce the user trust in the detector's predictions. A way to combat this is by allowing the detector to reject examples with high uncertainty (Learning to Reject). This requires employing a confidence metric that captures the distance to the decision boundary and setting a rejection threshold to reject low-confidence predictions. However, selecting a proper metric and setting the rejection threshold without labels are challenging tasks. In this paper, we solve these challenges by setting a constant rejection threshold on the stability metric computed by EXCEED. Our insight relies on a theoretical analysis of this metric. Moreover, setting a constant threshold results in strong guarantees: we estimate the test rejection rate, and derive a theoretical upper bound for both the rejection rate and the expected prediction cost. Experimentally, we show that our method outperforms some metric-based methods.

## 1 Introduction

Anomaly detection is the task of detecting unexpected behaviors in the data [6]. Often, these anomalies are critical adverse events such as the destruction or alteration of proprietary user data [29], water leaks in stores [50], breakdowns in gas [65] and wind [64] turbines, or failures in petroleum extraction [45]. Usually, anomalies are associated with a cost such as a monetary cost (e.g., maintenance, paying for fraudulent purchases) or a societal cost such as environmental damages (e.g., dispersion of petroleum or gas). Hence, detecting anomalies in a timely manner is an important problem.

When using an anomaly detector for decision-making, it is crucial that the user trusts the system. However, it is often hard or impossible to acquire labels for anomalies. Moreover, anomalies may not follow a pattern. Therefore anomaly detection is typically treated as an unsupervised learning problem where traditional algorithms learn a decision boundary by employing heuristics based on intuitions [22, 55, 61, 41], such as that anomalies are far away from normal examples [3]. Because these intuitions are hard to verify and may not hold in some cases, some predictions may have high uncertainty, especially for examples close to the decision boundary [32, 28]. As a result, the detector's predictions should be treated with some circumspection.

One way to increase user trust is to consider Learning to Reject [12]. In this setting, the model does not always make a prediction. Instead, it can abstain when it is at a heightened risk of making a mistake thereby improving its performance when it does offer a prediction. Abstention has the drawback that no prediction is made, which means that a person must intervene to make a decision. In the literature, two types of rejection have been identified [25]: novelty rejection allows the model to abstain when given an out-of-distribution (OOD) example, while ambiguity rejection enables abstention for a test example that is too close to the model's decision boundary. Because anomalies

37th Conference on Neural Information Processing Systems (NeurIPS 2023).

often are OOD examples, novelty rejection does not align well with our setting as the model would reject all OOD anomalies (i.e., a full class) [10, 63, 30]. On the other hand, current approaches for ambiguity rejection threshold what constitutes being too close to the decision boundary by evaluating the model's predictive performance on the examples for which it makes a prediction (i.e., accepted), and those where it abstains from making a prediction (i.e., rejected) [9, 44, 16]. Intuitively, the idea is to find a threshold where the model's predictive performance is (1) significantly lower on rejected examples than on accepted examples and (2) higher on accepted examples than on all examples (i.e., if it always makes a prediction). Unfortunately, existing learning to reject approaches that set a threshold in this manner require labeled data, which is not available in anomaly detection.

This paper fills this gap by proposing an approach to perform ambiguity rejection for anomaly detection in a completely unsupervised manner. Specifically, we make three major contributions. First, we conduct a thorough novel theoretical analysis of a stability metric for anomaly detection [49] and show that it has several previously unknown properties that are of great importance in the context of learning to reject. Namely, it captures the uncertainty close to the detector's decision boundary, and only limited number of examples get a stability value strictly lower than 1. Second, these enabls us to design an ambiguity rejection mechanism without *any labeled data* that offers strong guarantees which are often sought in Learning to Reject [12, 60, 7] We can derive an accurate estimate of the rejected examples proportion, as well as a theoretical upper bound that is satisfied with high probability. Moreover, given a cost function for different types of errors, we provide an estimated upper bound on the expected cost at the prediction time. Third, we evaluate our approach on an extensive set of unsupervised detectors and benchmark datasets and conclude that (1) it performs better than several adapted baselines based on other unsupervised metrics, and (2) our theoretical results hold in practice.

## 2   Preliminaries and notation

We will introduce the relevant background on anomaly detection, learning to reject, and the EXCEED's metric that this paper builds upon.

**Anomaly Detection.** Let $\mathcal{X}$ be a $d$ dimensional input space and $D = \{x_1, \ldots, x_n\}$ be a training set, where each $x_i \in \mathcal{X}$. The goal in anomaly detection is to train a detector $f \colon \mathcal{X} \to \mathbb{R}$ that maps examples to a real-valued anomaly score, denoted by $s$. In practice, it is necessary to convert these soft scores to a hard prediction, which requires setting a threshold $\lambda$. Assuming that higher scores equate to being more anomalous, a predicted label $\hat{y}$ can be made for an example $x$ as follows: $\hat{y} = 1$ (anomaly) if $s = f(x) \geq \lambda$, while $\hat{y} = 0$ (normal) if $s = f(x) < \lambda$. We let $\hat{Y}$ be the random variable that denotes the predicted label. Because of the absence of labels, one usually sets the threshold such that $\gamma \times n$ scores are $\geq \lambda$, where $\gamma$ is the dataset's contamination factor (i.e., expected proportion of anomalies) [51, 50].

**Learning to Reject.** Learning to reject extends the output space of the model to include the symbol ®, which means that the model abstains from making a prediction. This entails learning a second model $r$ (the rejector) to determine when the model abstains. A canonical example of ambiguity rejection is when $r$ consists of a pair [confidence $\mathcal{M}_s$, rejection threshold $\tau$] such that an example is rejected if the detector's confidence is lower than the threshold. The model output becomes

$$\hat{y}_{®} = \begin{cases} \hat{y} & \text{if } \mathcal{M}_s > \tau; \\ ® & \text{if } \mathcal{M}_s \leq \tau; \end{cases} \qquad \hat{y}_{®} \in \{0, 1, ®\}.$$

A standard approach is to evaluate different values for $\tau$ to find a balance between making too many incorrect predictions because $\tau$ is too low (i.e., $\hat{y} \neq y$ but $\mathcal{M}_s > \tau$) and rejecting correct predictions because $\tau$ is too high (i.e., $\hat{y} = y$ but $\mathcal{M}_s \leq \tau$) [9, 44, 16]. Unfortunately, in an unsupervised setting, it is impossible to evaluate the threshold because it relies on having access to labeled data.

**EXCEED's metric.** Traditional confidence metrics (such as calibrated class probabilities) quantify how likely a prediction is to be correct, This obviously requires labels [9] which are unavailable in an unsupervised setting. Thus, one option is to move the focus towards the **concept of stability**: given a fixed test example $x$ with anomaly score $s$, *perturbing the training data alters the model learning, which, in turn, affects the label prediction.* Intuitively, the more stable a detector's output is for a test example, the less sensitive its predicted label is to changes in the training data. On the other hand, when $\mathbb{P}(\hat{Y} = 1|s) \approx \mathbb{P}(\hat{Y} = 0|s) \approx 0.5$ the prediction for $x$ is highly unstable, as training the

detector with slightly different examples would flip its prediction for the same test score $s$. Thus, a stability-based confidence metric $\mathcal{M}_s$ can be expressed as the margin between the two classes' probabilities:

$$\mathcal{M}_s = |\mathbb{P}(\hat{Y} = 1|s) - \mathbb{P}(\hat{Y} = 0|s)| = |2\mathbb{P}(\hat{Y} = 1|s) - 1|,$$

where the lower $\mathcal{M}_s$ the more unstable the prediction.

Recently, Perini et al. introduced ExCeeD to estimate the detector's stability $\mathbb{P}(\hat{Y} = 1|s)$. Roughly speaking, ExCeeD uses a Bayesian formulation that simulates bootstrapping the training set as a form of perturbation. Formally, it measures such stability for a test score $s$ in two steps.

**First**, it computes the *training frequency* $\psi_n = \frac{|\{i \leq n : s_i \leq s\}|}{n} \in [0,1]$, i.e. the proportion of training scores lower than $s$. This expresses how extreme the score $s$ ranks with respect to the training scores. **Second**, it computes the probability that the score $s$ will be predicted as an anomaly when randomly drawing a training set of $n$ scores from the population of scores. In practice, this is the probability that the chosen threshold $\lambda$ will be less than or equal to the score $s$. The stability is therefore estimated as

$$\mathbb{P}(\hat{Y} = 1|s) = \sum_{i=n(1-\gamma)+1}^{n} \binom{n}{i} \left( \frac{1 + n\psi_n}{2 + n} \right)^i \left( \frac{n(1 - \psi_n) + 1}{2 + n} \right)^{n-i}. \tag{1}$$

*Assumption.* ExCeeD's Bayesian formulation requires assuming that $Y|s$ follows a Bernoulli distribution with parameter $p_s = \mathbb{P}(S \leq s)$, where $S$ is the detector's population of scores. Note that the stability metric is a detector property and, therefore, is tied to the specific choice of the unsupervised detector $f$.

## 3  Methodology

This paper addresses the following problem:

***Given:*** An unlabeled dataset $D$ with contamination $\gamma$, an unsupervised detector $f$, a cost function $c$;

***Do:*** Introduce a reject option to $f$, i.e. find a pair (confidence, threshold) that minimizes the cost.

We propose an anomaly detector-agnostic approach for performing learning to reject that requires *no labels*. Our key contribution is a novel theoretical analysis of the ExCeeD confidence metric that proves that *only a limited number of examples have confidence lower than* $1 - \varepsilon$ (Sec. 3.1). Intuitively, the detector's predictions for most examples would not be affected by slight perturbations of the training set: it is easy to identify the majority of normal examples and anomalies because they will strongly adhere to the data-driven heuristics that unsupervised anomaly detectors use. For example, using the data density as a measure of anomalousness [5] tends to identify all densely clustered normals and isolated anomalies, which constitute the majority of all examples. In contrast, only relatively few cases would be ambiguous and hence receive low confidence (e.g., small clusters of anomalies and normals at the edges of dense clusters).

Our approach is called **REJEX** (Rejecting via ExCeeD) and simply computes the stability-based confidence metric $\mathcal{M}_s$ and rejects any example with confidence that falls below threshold $\tau = 1 - \varepsilon$. Theoretically, this constant reject threshold provides several relevant guarantees. First, one often needs to control the proportion of rejections (namely, the *rejection rate*) to estimate the number of decisions left to the user. Thus, we propose *an estimator that only uses training instances to estimate the rejection rate at test time*. Second, because in some applications avoiding the risk of rejecting all the examples is a strict constraint, we provided *an upper bound for the rejection rate* (Sec. 3.2). Finally, we compute a *theoretical upper bound for a given cost function* that guarantees that using REJEX keeps the expected cost per example at test time low (Sec. 3.3).

### 3.1  Setting the Rejection Threshold through a Novel Theoretical Analysis of ExCeeD

Our novel theoretical analysis proves (1) that the stability metric by ExCeeD is lower than $1 - \varepsilon$ for a limited number of examples (Theorem 3.1), and (2) that such examples with low confidence are the ones close to the decision boundary (Corollay 3.2). Thus, we propose to reject all these uncertain examples by setting a rejection threshold

$$\tau = 1 - \varepsilon = 1 - 2e^{-T} \quad \text{for } T \geq 4,$$

where $2e^{-T}$ is the tolerance that excludes unlikely scenarios, and $T \geq 4$ is required for Theorem 3.1.

We motivate our approach as follows. Given an example $x$ with score $s$ and the proportion of lower training scores $\psi_n$, Theorem 3.1 shows that the confidence $\mathscr{M}_s$ is lower than $1 - 2e^{-T}$ (for $T \geq 4$) if $\psi_n$ belongs to the interval $[t_1, t_2]$. By analyzing $[t_1, t_2]$, Corollary 3.2 proves that the closer an example is to the decision boundary, the lower the confidence $\mathscr{M}_s$, and that a score $s = \lambda$ (decision threshold) has confidence $\mathscr{M}_s = 0$.

*Remark.* Perini et al. performed an asymptotic analysis of EXCEED that investigates the metric's behavior when the training set's size $n \to +\infty$. In contrast, our novel analysis is finite-sample and hence provides more practical insights, as real-world scenarios involve having a finite dataset with size $n \in \mathbb{N}$.

**Theorem 3.1** (Analysis of EXCEED). *Let $s$ be an anomaly score, and $\psi_n \in [0, 1]$ its training frequency. For $T \geq 4$, there exist $t_1 = t_1(n, \gamma, T) \in [0, 1]$, $t_2 = t_2(n, \gamma, T) \in [0, 1]$ such that*

$$\psi_n \in [t_1, t_2] \implies \mathscr{M}_s \leq 1 - 2e^{-T}.$$

*Proof.* See the Supplement for the formal proof. $\square$

The interval $[t_1, t_2]$ has two relevant properties. First, it becomes *narrower when increasing $n$* (P1) and *larger when increasing $T$* (P2). This means that collecting more training data results in smaller rejection regions while decreasing the tolerance $\varepsilon = 2e^{-T}$ has the opposite effect. Second, it is centered (not symmetrically) on $1 - \gamma$ (P3-P4), which means that *examples with anomaly scores close to the decision threshold $\lambda$ are the ones with a low confidence score* (P5). The next Corollary lists these properties.

**Corollary 3.2.** *Given $t_1, t_2$ as in Theorem 3.1, the following properties hold for any $s, n, \gamma, T \geq 4$:*

P1. $\lim_{n \to +\infty} t_1 = \lim_{n \to +\infty} t_2 = 1 - \gamma$;

P2. $t_1$ *and* $t_2$ *are, respectively, monotonic decreasing and increasing as functions of* $T$;

P3. *the interval always contains* $1 - \gamma$, *i.e.* $t_1 \leq 1 - \gamma \leq t_2$;

P4. *for* $n \to \infty$, *there exists* $s^*$ *with* $\psi_n = t^* \in [t_1, t_2]$ *such that* $t^* \to 1 - \gamma$ *and* $\mathscr{M}_s \to 0$.

P5. $\psi_n \in [t_1, t_2]$ *iff* $s \in [\lambda - u_1, \lambda + u_2]$, *where* $u_1(n, \gamma, T), u_2(n, \gamma, T)$ *are positive functions.*

*Proof sketch.* For P1, it is enough to observe that $t_1, t_2 \to 1 - \gamma$ for $n \to +\infty$. For P2 and P3, the result comes from simple algebraic steps. P4 follows from the surjectivity of $\mathscr{M}_s$ when $n \to +\infty$, the monotonicity of $\mathbb{P}(\hat{Y} = 1|s)$, from P1 with the squeeze theorem. Finally, P5 follows from $\psi_n \in [t_1, t_2] \implies s \in [\psi_n^{-1}(t_1), \psi_n^{-1}(t_2)]$, as $\psi_n$ is monotonic increasing, where $\psi_n^{-1}$ is the inverse-image of $\psi_n$. Because for P3 $1 - \gamma \in [t_1, t_2]$, it holds that $\psi_n^{-1}(t_1) \leq \psi_n^{-1}(1 - \gamma) = \lambda \leq \psi_n^{-1}(t_2)$. This implies that $s \in [\lambda - u_1, \lambda + u_2]$, where $u_1 = \lambda - \psi_n^{-1}(t_1)$, $u_2 = \lambda - \psi_n^{-1}(t_2)$. $\square$

### 3.2 Estimating and Bounding the Rejection Rate

It is important to have an estimate of the rejection rate, which is the proportion of examples for which the model will abstain from making a prediction. This is an important performance characteristic for differentiating among candidate models. Moreover, it is important that not all examples are rejected because such a model is useless in practice. We propose a way to estimate the rejection rate and Theorem 3.5 shows that our estimate approaches the true rate for large training sets. We strengthen our analysis and introduce an upper bound for the rejection rate, which guarantees that, with arbitrarily high probability, the rejection rate is kept lower than a constant (Theorem 3.6).

**Definition 3.3** (Rejection rate). Given the confidence metric $\mathscr{M}_s$ and the rejection threshold $\tau$, the *rejection rate* $\mathcal{R} = \mathbb{P}(\mathscr{M}_s \leq \tau)$ is the probability that a test example with score $s$ gets rejected.

We propose the following estimator for the reject rate:

**Definition 3.4** (Rejection rate estimator). Given anomaly scores $s$ with training frequencies $\psi_n$, let $g \colon [0, 1] \to [0, 1]$ be the function such that $\mathbb{P}(\hat{Y} = 1|s) = g(\psi_n)$ (see Eq. 1). We define the *rejection rate estimator* $\hat{\mathcal{R}}$ as

$$\hat{\mathcal{R}} = \hat{F}_{\psi_n}\left(g^{-1}\left(1 - e^{-T}\right)\right) - \hat{F}_{\psi_n}\left(g^{-1}\left(e^{-T}\right)\right) \tag{2}$$

where $g^{-1}$ is the inverse-image through $g$, and, for $u \in [0, 1]$, $\hat{F}_{\psi_n}(u) = \frac{|i \leq n \colon \psi_n(s_i) \leq u|}{n}$ is the empirical cumulative distribution of $\psi_n$.

Note that $\hat{\mathcal{R}}$ can be computed in practice, as the $\psi_n$ has a distribution that is arbitrarily close to uniform, as stated by Theorem A.1 and A.2 in the Supplement.

**Theorem 3.5** (Rejection rate estimate). *Let $g$ be as in Def. 3.4. Then, for high values of $n$, $\hat{\mathcal{R}} \approx \mathcal{R}$.*

*Proof.* From the definition of rejection rate 3.3, it follows

$$\mathcal{R} = \mathbb{P}\left(\mathscr{M}_s \leq 1 - 2e^{-T}\right) = \mathbb{P}\left(\mathbb{P}(\hat{Y}=1|s) \in \left[e^{-T}, 1 - e^{-T}\right]\right) = \mathbb{P}\left(g(\psi_n) \in \left[e^{-T}, 1 - e^{-T}\right]\right)$$

$$= \mathbb{P}\left(\psi_n \in \left[g^{-1}\left(e^{-T}\right), g^{-1}\left(1 - e^{-T}\right)\right]\right) = F_{\psi_n}\left(g^{-1}\left(1 - e^{-T}\right)\right) - F_{\psi_n}\left(g^{-1}\left(e^{-T}\right)\right).$$

where $F_{\psi_n}(\cdot) = \mathbb{P}(\psi_n \leq \cdot)$ is the theoretical cumulative distribution of $\psi_n$. Because the true distribution of $\psi_n$ for test examples is unknown, the estimator approximates $F_{\psi_n}$ using the training scores $s_i$ and computes the empirical $\hat{F}_{\psi_n}$. As a result,

$$\mathcal{R} \approx \hat{F}_{\psi_n}\left(g^{-1}\left(1 - e^{-T}\right)\right) - \hat{F}_{\psi_n}\left(g^{-1}\left(e^{-T}\right)\right) = \hat{\mathcal{R}}.$$

$\square$

**Theorem 3.6** (Rejection rate upper bound). *Let $s$ be an anomaly score, $\mathscr{M}_s$ be its confidence value, and $\tau = 1 - 2e^{-T}$ be the rejection threshold. For $n \in \mathbb{N}$, $\gamma \in [0, 0.5)$, and small $\delta > 0$, there exists a positive real function $h(n, \gamma, T, \delta)$ such that $\mathcal{R} \leq h(n, \gamma, T, \delta)$ with probability at least $1 - \delta$, i.e. the rejection rate is bounded.*

*Proof.* Theorem 3.1 states that there exists two functions $t_1 = t_1(n, \gamma, T), t_2 = t_2(n, \gamma, T) \in [0, 1]$ such that the confidence is lower than $\tau$ if $\psi_n \in [t_1, t_2]$. Moreover, Theorems A.1 and A.2 claim that $\psi_n$ has a distribution that is close to uniform with high probability (see the theorems and proofs in the Supplement). As a result, with probability at least $1 - \delta$, we find $h(n, \gamma, T, \delta)$ as follows:

$$\mathcal{R} = \mathbb{P}(\mathscr{M}_s \leq 1 - 2e^{-T}) \overset{\text{T3.1}}{\leq} \mathbb{P}\left(\psi_n \in [t_1, t_2]\right) = F_{\psi_n}(t_2) - F_{\psi_n}(t_1)$$

$$\overset{\text{T A.2}}{\leq} F_{\psi}(t_2) - F_{\psi}(t_1) + 2\sqrt{\frac{\ln \frac{2}{\delta}}{2n}} \overset{\text{T A.1}}{=} t_2(n, \gamma, T) - t_1(n, \gamma, T) + 2\sqrt{\frac{\ln \frac{2}{\delta}}{2n}} = h(n, \gamma, T, \delta).$$

$\square$

## 3.3 Upper Bounding the Expected Test Time Cost

In a learning with reject scenario, there are costs associated with three outcomes: false positives ($c_{fp} > 0$), false negatives ($c_{fn} > 0$), and rejection ($c_r$) because abstaining typically involves having a person intervene. Estimating an expected per example prediction cost at test time can help with model selection and give a sense of performance. Theorem 3.8 provides an upper bound on the expected per example cost when (1) using our estimated rejection rate (Theorem 3.5), and (2) setting the decision threshold $\lambda$ as in Sec. 2.

**Definition 3.7** (Cost function). Let $Y$ be the true label random variable. Given the costs $c_{fp} > 0$, $c_{fn} > 0$, and $c_r$, the **cost function** is a function $c \colon \{0, 1\} \times \{0, 1, \circledR\} \to \mathbb{R}$ such that

$$c(Y, \hat{Y}) = c_r\mathbb{P}(\hat{Y}=\circledR) + c_{fp}\mathbb{P}(\hat{Y}=1|Y=0) + c_{fn}\mathbb{P}(\hat{Y}=0|Y=1)$$

Note that defining a specific cost function requires domain knowledge. Following the learning to reject literature, we set an additive cost function. Moreover, the rejection cost needs to satisfy the inequality $c_r \leq \min\{(1-\gamma)c_{fp}, \gamma c_{fn}\}$. This avoids the possibility of predicting always anomaly for an expected cost of $(1 - \gamma)c_{fp}$, or always normal with an expected cost of $\gamma c_{fn}$ [52].

**Theorem 3.8.** *Let $c$ be a cost function as defined in Def. 3.7, and $g$ be as in Def. 3.4. Given a (test) example $x$ with score $s$, the expected example-wise cost is bounded by*

$$\mathbb{E}_x[c] \leq \min\{\gamma, A\}c_{fn} + (1 - B)c_{fp} + (B - A)c_r, \tag{3}$$

*where $A = \hat{F}_{\psi_n}(g^{-1}(e^{-T}))$ and $B = \hat{F}_{\psi_n}(g^{-1}(1 - e^{-T}))$ are as in Theorem 3.5.*

*Proof.* We indicate the true label random variable as $Y$, and the non-rejected false positives and false negatives as, respectively,

$$FP = \mathbb{P}\left(\hat{Y} = 1 | Y = 0, \mathscr{M}_s > 1 - 2e^{-T}\right) \quad FN = \mathbb{P}\left(\hat{Y} = 0 | Y = 1, \mathscr{M}_s > 1 - 2e^{-T}\right)$$

Using Theorem 3.5 results in

$$\mathbb{E}_x[c] = \mathbb{E}_x[c_{fn}FN + c_{fp}FP + c_r\mathcal{R}] = \mathbb{E}_x[c_{fn}FN] + \mathbb{E}_x[c_{fp}FP] + c_r(B - A)$$

where $A = \hat{F}_{\psi_n}(g^{-1}\left(e^{-T}\right))$, $B = \hat{F}_{\psi_n}(g^{-1}\left(1 - e^{-T}\right))$ come from Theorem 3.5. Now we observe that setting a decision threshold $\lambda$ such that $n \times \gamma$ scores are higher implies that, on expectation, the detector predicts a proportion of positives equal to $\gamma = \mathbb{P}(Y = 1)$. Moreover, for $\varepsilon = 2e^{-T}$,

- $FP \leq \mathbb{P}\left(\hat{Y} = 1 | \mathscr{M}_s > 1 - \varepsilon\right) = 1 - B$ as false positives must be less than total accepted positive predictions;

- $FN \leq \gamma$ and $FN \leq \mathbb{P}\left(\hat{Y} = 0 | \mathscr{M}_s > 1 - \varepsilon\right) = A$, as you cannot have more false negatives than positives ($\gamma$), nor than accepted negative predictions ($A$).

From these observations, we conclude that $\mathbb{E}_x[c] \leq \min\{\gamma, A\}c_{fn} + (1 - B)c_{fp} + (B - A)c_r$. $\square$

## 4 Related work

There is no research on learning to reject in unsupervised anomaly detection. However, **three** main research lines are connected to this work.

**1) Supervised methods.** If some labels are available, one can use traditional supervised approaches to add the reject option into the detector [11, 38]. Commonly, labels can be used to find the optimal rejection threshold in two ways: 1) by trading off the model performance (e.g., AUC) on the accepted examples with its rejection rate [24, 1], or 2) by minimizing a cost function [46, 7], a risk function [18, 27], or an error function [35, 33]. Alternatively, one can include the reject option in the model and directly optimize it during the learning phase [60, 12, 31].

**2) Self-Supervised methods.** If labels are not available, one can leverage self-supervised approaches to generate pseudo-labels in order to apply traditional supervised learning to reject methods [26, 59, 19, 37]. For example, one can employ any unsupervised anomaly detector to assign training labels, fit a (semi-)supervised detector (such as DEEPSAD [57] or REPEN [47]) on the pseudo labels, compute a confidence metric [14], and find the optimal rejection threshold by minimizing the cost function treating the pseudo-labels as the ground truth.

**3) Optimizing unsupervised metrics.** There exist several unsupervised metrics (i.e., they can be computed without labels) for quantifying detector quality [43]. Because they do not need labels, one can find the rejection threshold by maximizing the margin between the detector's quality (computed using such metric) on the accepted and on the rejected examples [54]. This allows us to obtain a model that performs well on the accepted examples and poorly on the rejected ones, which is exactly the same intuition that underlies the supervised approaches. Some examples of existing unsupervised metrics (see [43]) are the following. EM and MV [20] quantify the clusterness of inlier scores, where more compact scores indicate better models. STABILITY [48] measures the robustness of anomaly detectors' predictions by looking at how consistently they rank examples by anomalousness. UDR [15] is a model-selection metric that selects the model with a hyperparameter setting that yields consistent results across various seeds, which can be used to set the rejection threshold through the analogy [hyperparameter, seed] and [rejection threshold, detectors]. Finally, ENS [56, 67] measures the detector trustworthiness as the ranking-based similarity (e.g., correlation) of a detector's output to the "pseudo ground truth", computed via aggregating the output of an ensemble of detectors, which allows one to set the rejection threshold that maximizes the correlation between the detector's and the ensemble's outputs.

# 5 Experiments

We experimentally address the following research questions:

**Q1.** How does REJEX's cost compare to the baselines?

**Q2.** How does varying the cost function affect the results?

**Q3.** How does REJEX's CPU time compare to the baselines?

**Q4.** Do the theoretical results hold in practice?

**Q5.** Would REJEX's performance significantly improve if it had access to training labels?

## 5.1 Experimental Setup

**Methods.** We compare **REJEX**[1] against 7 baselines for setting the rejection threshold. These can be divided into three categories: no rejection, self-supervised, and unsupervised metric based.

We use one method **NOREJECT** that always makes predictions and never rejects (no reject option).

We consider one self-supervised approach **SS-REPEN** [47]. This uses (any) unsupervised detector to obtain pseudo labels for the training set. It then sets the rejection threshold as follows: 1) it creates a held-out validation set (20%), 2) it fits REPEN, a state-of-the-art (semi-)supervised anomaly detector on the training set with the pseudo labels, 3) it computes on the validation set the confidence values as the margin between REPEN's predicted class probabilities $|\mathbb{P}(Y = 1|s) - \mathbb{P}(Y = 0|s)|$, 4) it finds the optimal threshold $\tau$ by minimizing the total cost obtained on the validation set.

We consider 5 approaches that employ an existing unsupervised metric to set the rejection threshold and hence do not require having access to labels. **MV** [20], **EM** [20], and **STABILITY** [48] are unsupervised metric-based methods based on stand-alone internal evaluations that use a single anomaly detector to measure its quality, **UDR** [15] and **ENS** [56] are unsupervised consensus-based metrics that an ensemble of detectors (all 12 considered in our experiments) to measure a detector's quality.[2] We apply each of these 5 baselines as follows. 1) We apply the unsupervised detector to assign an anomaly score to each train set example. 2) We convert these scores into class probabilities using [34]. 3) We compute the confidence scores on the training set as difference between these probabilities: $|\mathbb{P}(Y = 1|s) - \mathbb{P}(Y = 0|s)|$. 4) We evaluate possible thresholds on this confidence by computing the considered unsupervised metric on the accepted and on the rejected examples and select the threshold that maximizes the difference in the metric's value on these two sets of examples. This aligns with the common learning to reject criteria for picking a threshold [9, 54] such that the model performs well on the accepted examples and poorly on the rejected ones.

**Data.** We carry out our study on 34 publicly available benchmark datasets, widely used in the literature [23]. These datasets cover many application domains, including healthcare (e.g., disease diagnosis), audio and language processing (e.g., speech recognition), image processing (e.g., object identification), and finance (e.g., fraud detection). To limit the computational time, we randomly sub-sample 20,000 examples from all large datasets. Table 3 in the Supplement provides further details.

**Anomaly Detectors and Hyperparameters.** We set our tolerance $\varepsilon = 2e^{-T}$ with $T = 32$. Note that the exponential smooths out the effect of $T \geq 4$, which makes setting a different $T$ have little impact. We use a set of 12 unsupervised anomaly detectors implemented in PYOD [66] with default hyperparameters [62] because the unsupervised setting does not allow us to tune them: KNN [3], IFOREST [42], LOF [5], OCSVM [58], AE [8], HBOS [21], LODA [53], COPOD [39], GMM [2], ECOD [40], KDE [36], INNE [4]. We set all the baselines' rejection threshold via Bayesian Optimization with 50 calls [17].

**Setup.** For each [dataset, detector] pair, we proceed as follows: (1) we split the dataset into training and test sets (80-20) using 5 fold cross-validation; (2) we use the detector to assign the anomaly scores on the training set; (3) we use either REJEX or a baseline to set the rejection threshold;

---

[1]Code available at: `https://github.com/Lorenzo-Perini/RejEx`.

[2]Sec. 4 describes these approaches.

(4) we measure the total cost on the test set using the given cost function. We carry out a total of $34 \times 12 \times 5 = 2040$ experiments. All experiments were run on an Intel(R) Xeon(R) Silver 4214 CPU.

## 5.2 Experimental Results

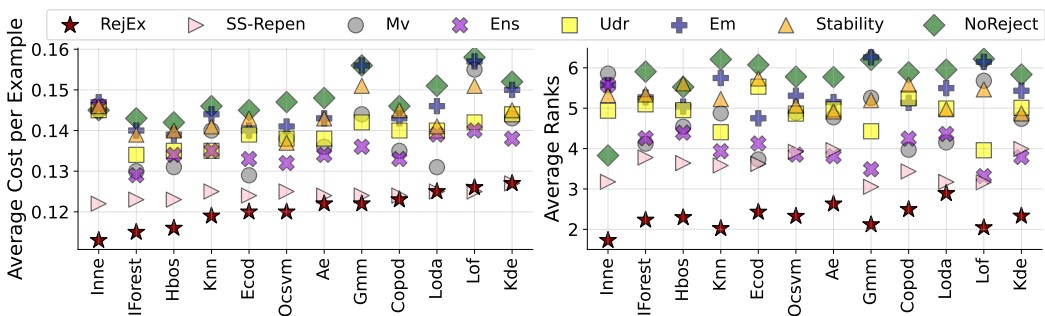

Figure 1: Average cost per example (left) and rank (right) aggregated per detector (x-axis) over all the datasets. Our method obtains the lowest (best) cost for 9 out of 12 detectors and it always has the lowest (best) ranking position for $c_{fp} = c_{fn} = 1$, $c_r = \gamma$.

**Q1: REJEX against the baselines.** Figure 1 shows the comparison between our method and the baselines, grouped by detector, when setting the costs $c_{fp} = c_{fn} = 1$ and $c_r = \gamma$ (see the Supplement for further details). REJEX achieves the lowest (best) cost per example for 9 out of 12 detectors (left-hand side) and similar values to SS-REPEN when using LODA, LOF and KDE. Averaging over the detectors, REJEX reduces the relative cost by more than $5\%$ vs SS-REPEN, $11\%$ vs ENS, $13\%$ vs MV and UDR, $17\%$ vs EM, $19\%$ vs NOREJECT. Table 4 (Supplement) shows a detailed breakdown.

For each experiment, we rank all the methods from 1 to 8, where position 1 indicates the lowest (best) cost. The right-hand side of Figure 1 shows that REJEX always obtains the lowest average ranking. We run a statistical analysis separately for each detector: the Friedman test rejects the null-hypothesis that all methods perform similarly (p-value $< e^{-16}$) for all the detectors. The ranking-based post-hoc Bonferroni-Dunn statistical test [13] with $\alpha = 0.05$ finds that REJEX is significantly better than the baselines for 6 detectors (INNE, IFOREST, HBOS, KNN, ECOD, OCSVM).

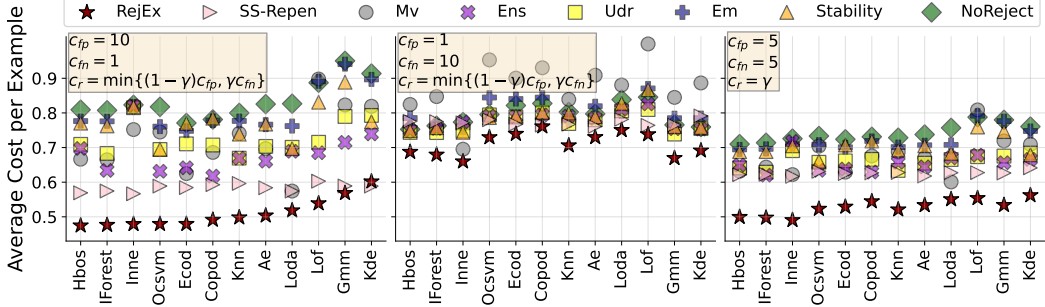

Figure 2: Average cost per example aggregated by detector over the $34$ datasets when varying the three costs on three representative cases: (left) false positives are penalized more, (center) false negatives are penalized more, (right) rejection has a lower cost than FPs and FNs.

**Q2. Varying the costs** $c_{fp}$, $c_{fn}$, $c_r$**.** The three costs $c_{fp}$, $c_{fn}$, and $c_r$ are usually set based on domain knowledge: whether to penalize the false positives or the false negatives more depends on the application domain. Moreover, the rejection cost needs to satisfy the constraint $c_r \leq \min\{(1 -$

$\gamma)c_{fp}, \gamma c_{fn}\}$ [52]. Therefore, we study their impact on three representative cases: (case 1) high false positive cost ($c_{fp} = 10$, $c_{fn} = 1$, $c_r = \min\{10(1 - \gamma), \gamma)$, (case 2) high false negative cost ($c_{fp} = 1$, $c_{fn} = 10$, $c_r = \min\{(1 - \gamma), 10\gamma)$, and (case 3) same cost for both mispredictions but low rejection cost ($c_{fp} = 5$, $c_{fn} = 5$, $c_r = \gamma$). Note that scaling all the costs has no effect on the relative comparison between the methods, so the last case is equivalent to $c_{fp} = 1$, $c_{fn} = 1$, and $c_r = \gamma/5$.

Figure 2 shows results for the three scenarios. Compared to the unsupervised metric-based methods, the left plot shows that our method is clearly the best for high false positives cost: for 11 out of 12 detectors, REJEX obtains both the lowest (or similar for GMM) average cost and the lowest average ranking position. This indicates that using REJEX is suitable when false alarms are expensive. Similarly, the right plot illustrates that REJEX outperforms all the baselines for all the detectors when the rejection cost is low (w.r.t. the false positive and false negative costs). Even when the false negative cost is high (central plot), REJEX obtains the lowest average cost for 11 detectors and has always the lowest average rank per detector. See the Supplement (Table 6 and 7) for more details.

Table 1: Average CPU time (in ms) per training example ($\pm$ std) to set the rejection threshold aggregated over all the datasets when using IFOREST, HBOS, and COPOD as unsupervised anomaly detector. REJEX has a lower time than all the methods but NOREJECT, which uses no reject option.

| | | | | CPU time in ms (mean $\pm$ std.) | | | | |
|---|---|---|---|---|---|---|---|---|
| DETECTOR | NOREJECT | **REJEX** | SS-REPEN | MV | EM | UDR | ENS | STABILITY |
| IFOREST | 0.0$\pm$0.0 | **0.06$\pm$0.22** | 90$\pm$68 | 89$\pm$128 | 155$\pm$161 | 120$\pm$132 | 122$\pm$135 | 916$\pm$900 |
| HBOS | 0.0$\pm$0.0 | **0.13$\pm$0.93** | 89$\pm$53 | 39$\pm$81 | 80$\pm$129 | 200$\pm$338 | 210$\pm$358 | 142$\pm$242 |
| COPOD | 0.0$\pm$0.0 | **0.04$\pm$0.04** | 84$\pm$53 | 21$\pm$28 | 81$\pm$60 | 119$\pm$131 | 123$\pm$138 | 140$\pm$248 |

**Q3. Comparing the CPU time.** Table 1 reports CPU time in milliseconds per training example aggregated over the 34 datasets needed for each method to set the rejection threshold on three unsupervised anomaly detectors (IFOREST, HBOS, COPOD). NOREJECT has CPU time equal to 0 because it does not use any reject option. REJEX takes just a little more time than NOREJECT because computing EXCEED has linear time while setting a constant threshold has constant time. In contrast, all other methods take $1000\times$ longer because they evaluate multiple thresholds. For some of these (e.g., STABILITY), this involves an expensive internal procedure.

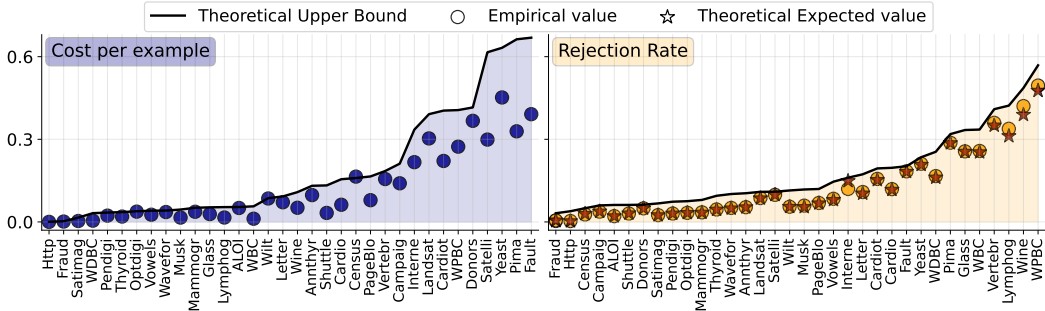

Figure 3: Average cost per example (left) and average rejection rate (right) at test time aggregated by dataset over the 12 detectors. In both plots, the empirical value (circle) is always lower than the predicted upper bound (continuous black line), which makes it consistent with the theory. On the right, the expected rejection rates (stars) are almost identical to the empirical values.

**Q4. Checking on the theoretical results.** Section 3 introduces three theoretical results: the rejection rate estimate (Theorem 3.5), and the upper bound for the rejection rate (Theorem 3.6) and for the cost (Theorem 3.8). We run experiments to verify whether they hold in practice. Figure 3 shows the results aggregated over the detectors. The left-hand side confirms that the prediction cost per example (blue circle) is always $\leq$ than the upper bound (black line). Note that the upper bound is sufficiently strict, as in some cases it equals the empirical cost (e.g., Census, Wilt, Optdigits).

The right-hand side shows that our rejection rate estimate (orange star) is almost identical to the empirical rejection rate (orange circle) for most of the datasets, especially the large ones. On the other hand, small datasets have the largest gap, e.g., Wine ($n = 129$), Lymphography ($n = 148$), WPBC ($n = 198$), Vertebral ($n = 240$). Finally, the empirical rejection rate is always lower than the theoretical upper bound (black line), which we compute by using the empirical frequencies $\psi_n$.

Table 2: Mean $\pm$ std. for the **cost per example** (on the left) and the **rejection rate** (on the right) at test time on a per detector basis and aggregated over the datasets.

| | COST PER EXAMPLE (MEAN $\pm$ STD.) | | REJECTION RATE (MEAN $\pm$ STD.) | |
| DETECTOR | REJEX | ORACLE | REJEX | ORACLE |
|---|---|---|---|---|
| AE | $0.126 \pm 0.139$ | $0.126 \pm 0.139$ | $0.131 \pm 0.132$ | $0.118 \pm 0.125$ |
| COPOD | $0.123 \pm 0.140$ | $0.121 \pm 0.140$ | $0.123 \pm 0.131$ | $0.101 \pm 0.114$ |
| ECOD | $0.119 \pm 0.138$ | $0.118 \pm 0.138$ | $0.125 \pm 0.130$ | $0.107 \pm 0.114$ |
| GMM | $0.123 \pm 0.135$ | $0.122 \pm 0.134$ | $0.139 \pm 0.143$ | $0.132 \pm 0.136$ |
| HBOS | $0.118 \pm 0.129$ | $0.118 \pm 0.129$ | $0.139 \pm 0.148$ | $0.114 \pm 0.128$ |
| IFOREST | $0.118 \pm 0.129$ | $0.118 \pm 0.128$ | $0.127 \pm 0.131$ | $0.118 \pm 0.130$ |
| INNE | $0.115 \pm 0.129$ | $0.115 \pm 0.128$ | $0.132 \pm 0.132$ | $0.122 \pm 0.125$ |
| KDE | $0.129 \pm 0.140$ | $0.129 \pm 0.139$ | $0.121 \pm 0.129$ | $0.105 \pm 0.120$ |
| KNN | $0.119 \pm 0.123$ | $0.118 \pm 0.123$ | $0.127 \pm 0.129$ | $0.112 \pm 0.117$ |
| LODA | $0.125 \pm 0.133$ | $0.122 \pm 0.130$ | $0.126 \pm 0.124$ | $0.110 \pm 0.114$ |
| LOF | $0.126 \pm 0.131$ | $0.125 \pm 0.131$ | $0.129 \pm 0.126$ | $0.118 \pm 0.115$ |
| OCSVM | $0.120 \pm 0.131$ | $0.120 \pm 0.131$ | $0.126 \pm 0.128$ | $0.107 \pm 0.115$ |
| AVG. | $0.122 \pm 0.133$ | $0.121 \pm 0.133$ | $0.129 \pm 0.132$ | $0.114 \pm 0.121$ |

**Q5. Impact of training labels on REJEX.**   We simulate having access to the training labels and include an extra baseline: ORACLE uses EXCEED as a confidence metric and sets the (optimal) rejection threshold by minimizing the cost function using the training labels. Table 2 shows the average cost and rejection rates at test time obtained by the two methods. Overall, REJEX obtains an average cost that is only $0.6\%$ higher than ORACLE's cost. On a per-detector basis, REJEX obtains a $2.5\%$ higher cost in the worst case (with LODA), while getting only a $0.08\%$ increase in the best case (with KDE). Comparing the rejection rates, REJEX rejects on average only $\approx 1.5$ percentage points more examples than ORACLE ($12.9\%$ vs $11.4\%$). The supplement provides further details.

## 6    Conclusion and Limitations

This paper addressed learning to reject in the context of unsupervised anomaly detection. The key challenge was how to set the rejection threshold without access to labels which are required by all existing approaches We proposed an approach REJEX that exploits our novel theoretical analysis of the EXCEED confidence metric. Our new analysis shows that it is possible to set a constant rejection threshold and that doing so offers strong theoretical guarantees. First, we can estimate the proportion of rejected test examples and provide an upper bound for our estimate. Second, we can provide a theoretical upper bound on the expected test-time prediction cost per example. Experimentally, we compared REJEX against several (unsupervised) metric-based methods and showed that, for the majority of anomaly detectors, it obtained lower (better) cost. Moreover, we proved that our theoretical results hold in practice and that our rejection rate estimate is almost identical to the true value in the majority of cases.

**Limitations.**  Because REJEX does not rely on labels, it can only give a coarse-grained view of performance. For example, in many applications anomalies will have varying costs (i.e., there are instance-specific costs) which we cannot account for. Moreover, REJEX has a strictly positive rejection rate, which may increase the cost of a highly accurate detector. However, this happens only in $\approx 5\%$ of our experiments.

## Acknowledgements

This research is supported by an FB Ph.D. fellowship by FWO-Vlaanderen (grant 1166222N) [LP], the Flemish Government under the "Onderzoeksprogramma Artificiële Intelligentie (AI) Vlaanderen" programme [LP,JD], and KUL Research Fund iBOF/21/075 [JD].

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

# Supplement

In this supplementary material we (1) provide additional theorems and proofs for Section 3, and (2) further describe the experimental results.

## A Theoretical Results

Firstly, we provide the proof for Theorem 3.1.

**Theorem 3.1** (Analysis of EXCEED) Let $s$ be an anomaly score, and $\psi_n \in [0,1]$ the proportion of training scores $\leq s$. For $T \geq 4$, there exist $t_1 = t_1(n, \gamma, T) \in [0,1]$, $t_2 = t_2(n, \gamma, T) \in [0,1]$ such that

$$\psi_n \in [t_1, t_2] \implies \mathscr{M}_s \leq 1 - 2e^{-T}.$$

*Proof.* We split this proof into two parts: we show that the reverse inequalities, i.e. that **(a)** if $\psi_n \leq t_1$, then $\mathscr{M}_s \geq 1 - 2e^{-T}$, and **(b)** if $\psi_n \geq t_2$, then $\mathscr{M}_s \geq 1 - 2e^{-T}$, hold and prove the final statement because $\mathbb{P}(\hat{Y} = 1|s)$ is monotonic increasing on $s$.

**(a)** The probability $\mathbb{P}(\hat{Y} = 1|s)$ (as in Eq. 1) can be seen as the cumulative distribution $F$ of a binomial random variable $\mathcal{B}(q_s, n)$ with at most $n\gamma - 1$ successes out of $n$ trials, with $q_s = \frac{n(1-\psi_n)+1}{2+n}$ as the success probability. By applying Hoeffding's inequality, we obtain the upper bound

$$\mathbb{P}(\hat{Y} = 1|s) \leq \exp\left(-2n\left(\frac{n(1-\psi_n)+1}{2+n} - \frac{n\gamma-1}{n}\right)^2\right)$$

that holds for the constraint $\psi_n \leq \frac{2+n}{n^2} + \frac{1-2\gamma}{n} + (1-\gamma)$. Because $\mathbb{P}(\hat{Y} = 1|s) \leq e^{-T}$ implies that $\mathscr{M}_s \geq 1 - 2e^{-T}$, we search for the values of $\psi_n$ such that the upper bound is $\leq e^{-T}$. Forcing the upper bound $\leq e^{-T}$ results in

$$2n\left(\frac{n(1-\psi_n)+1}{2+n} - \frac{n\gamma-1}{n}\right)^2 - T \geq 0,$$

which is satisfied for $(I_1)$ $0 \leq \psi_n \leq A_1 - \sqrt{B_1}$ and $(I_2)$ $A_1 + \sqrt{B_1} \leq \psi_n \leq 1$, where

$$A_1 = \frac{2 + n(n+1)(1-\gamma)}{n^2} \qquad B_1 = \frac{2n\left(-3\gamma^2 - 2n(1-\gamma)^2 + 4\gamma - 3\right) + T(n+2)^2 - 8}{2n^3}.$$

However, for $T \geq 4$, no values of $n$, $\gamma$, and $T$ that satisfy the constraint on $\psi_n$ also satisfy $I_2$. Moving to $I_1$, we find out that if $\psi_n$ satisfies $I_1$, then it also satisfies the constraint on $\psi_n$ for any $n$, $\gamma$, and $T$. Therefore, we we set $t_1(n, \gamma, T) = A_1 - \sqrt{B_1}$. As a result,

$$\psi_n \leq t_1 \implies \mathbb{P}(\hat{Y} = 1|s) \leq e^{-T} \implies \mathscr{M}_s \geq 1 - 2e^{-T}.$$

**(b)** Similarly, $\mathbb{P}(\hat{Y} = 0|s)$ can be seen as the cumulative distribution $F$ of $\mathcal{B}(p_s, n)$, with $n(1-\gamma)$ successes and $p_s = \frac{1+n\psi_n(s)}{2+n}$. By seeing the binomial as a sum of Bernoulli random variables, and using the property of its cumulative distribution $F(n(1-\gamma), n, p_s) + F(n\gamma - 1, n, 1 - p_s) = 1$, we apply the Hoeffding's inequality and compare such upper bound to the $e^{-T}$. We obtain

$$2n\left(\frac{1+\psi_n n}{2+n} - (1-\gamma)\right)^2 - T \geq 0$$

that holds with the constraint $\psi_n \geq \frac{(2+n)(1-\gamma)-1}{n}$. The quadratic inequality in $\psi_n$ has solutions for $(I_1)$ $0 \leq \psi_n \leq A_2 - \sqrt{B_2}$ and $(I_2)$ $A_2 + \sqrt{B_2} \leq \psi_n \leq 1$, where $A_2 = \frac{(2+n)(1-\gamma)-1}{n}$, and $B_2 = \frac{T(n+2)^2}{2n^3}$. However, the constraint limits the solutions to $I_2$, i.e. for $\psi_n \geq A_2 + \sqrt{B_2}$. Thus, we set $t_2(n, \gamma, T) = A_2 + \sqrt{B_2}$ and conclude that

$$\psi_n \geq t_2 \implies \mathbb{P}(\hat{Y} = 1|s) \geq 1 - e^{-T} \implies \mathscr{M}_s \geq 1 - 2e^{-T}.$$

$\square$

Secondly, Theorem 3.6 relies on two important results: given $S$ the anomaly score random variable, (1) if $\psi_n$ was the *theoretical* cumulative of $S$, it would have a uniform distribution (Theorem A.1), but because in practice (2) $\psi_n$ is the *empirical* cumulative of $S$, its distribution is close to uniform with high probability (Theorem A.2). We prove these results in the following theorems.

**Theorem A.1.** *Let $S$ be the anomaly score random variable, and $\psi = F_S(S)$ be the cumulative distribution of $S$ applied to $S$ itself. Then $\psi \sim Unif(0,1)$.*

*Proof.* We prove that, if $\psi \sim Unif(0,1)$, then $F_\psi(t) = t$ for any $t \in [0,1]$:

$$F_\psi(t) = \mathbb{P}(\psi \leq t) = \mathbb{P}(F_S(S) \leq t) = \mathbb{P}(S \leq F_S^{-1}(t)) = F_S(F_S^{-1}(t)) = t \implies \psi \sim Unif(0,1).$$

$\square$

**Theorem A.2.** *Let $\psi$ be as in Theorem A.1, and $F_{\psi_n}$ be its empirical distribution obtained from a sample of size $n$. For any small $\delta > 0$ and $t \in [0,1]$, with probability $> 1 - \delta$*

$$F_{\psi_n}(t) \in \left[ F_\psi(t) - \sqrt{\frac{\ln\frac{2}{\delta}}{2n}}, F_\psi(t) + \sqrt{\frac{\ln\frac{2}{\delta}}{2n}} \right].$$

*Proof.* For any $\varepsilon > 0$, the DKW inequality implies

$$\mathbb{P}\left( \sup_{t\in[0,1]} |F_{\psi_n}(t) - F_\psi(t)| > \varepsilon \right) \leq 2\exp\left(-2n\varepsilon^2\right).$$

By setting $\delta = 2\exp\left(-2n\varepsilon^2\right)$, i.e. $\varepsilon = \sqrt{\frac{\ln\frac{2}{\delta}}{2n}}$, and using the complementary probability we conclude that

$$\mathbb{P}\left( \sup_{t\in[0,1]} |F_{\psi_n}(t) - F_\psi(t)| \leq \sqrt{\frac{\ln\frac{2}{\delta}}{2n}} \right) > 1 - \delta.$$

$\square$

# B  Experiments

**Data.** Table 3 shows the properties of the 34 datasets used for the experimental comparison, in terms of number of examples, features, and contamination factor $\gamma$. The datasets can be downloaded in the following link: `https://github.com/Minqi824/ADBench/tree/main/datasets/Classical`.

**Q1. REJEX against the baselines.** Table 4 and Table 5 show the results (mean $\pm$ std) aggregated by detectors in terms of, respectively, cost per example and ranking position. Results confirm that REJEX obtains an average cost per example lower than all the baselines for 9 out of 12 detectors, which is similar to the runner-up SS-REPEN for the remaining 3 detectors. Moreover, REJEX has always the best (lowest) average ranking position.

**Q2. Varying the costs $c_{fp}$, $c_{fn}$, $c_r$.** Table 6 and Table 7 show the average cost per example and the ranking position (mean $\pm$ std) aggregated by detectors for three representative cost functions, as discussed in the paper. Results are similar in all three cases. For high false positives cost ($c_{fp} = 10$), REJEX obtains an average cost per example lower than all the baselines for 11 out of 12 detectors and always the best average ranking position. For high false negative cost ($c_{fn} = 10$) as well as for low rejection cost ($c_{fp} = 5, c_{fn} = 5, c_r = \gamma$), it has the lowest average cost for all detectors and always the best average ranking. Moreover, when rejection is highly valuable (low cost), REJEX's cost has a large gap with respect to the baselines, which means that it is particularly useful when rejection is less expensive.

Table 3: Properties (number of examples, features, and contamination factor) of the 34 benchmark datasets used for the experiments.

| Dataset | #Examples | #Features | $\gamma$ |
|---|---|---|---|
| ALOI | 20000 | 27 | 0.0315 |
| ANNTHYROID | 7062 | 6 | 0.0756 |
| CAMPAIGN | 20000 | 62 | 0.1127 |
| CARDIO | 1822 | 21 | 0.0960 |
| CARDIOTOCOGRAPHY | 2110 | 21 | 0.2204 |
| CENSUS | 20000 | 500 | 0.0854 |
| DONORS | 20000 | 10 | 0.2146 |
| FAULT | 1941 | 27 | 0.3467 |
| FRAUD | 20000 | 29 | 0.0021 |
| GLASS | 213 | 7 | 0.0423 |
| HTTP | 20000 | 3 | 0.0004 |
| INTERNETADS | 1966 | 1555 | 0.1872 |
| LANDSAT | 6435 | 36 | 0.2071 |
| LETTER | 1598 | 32 | 0.0626 |
| LYMPHOGRAPHY | 148 | 18 | 0.0405 |
| MAMMOGRAPHY | 7848 | 6 | 0.0322 |
| MUSK | 3062 | 166 | 0.0317 |
| OPTDIGITS | 5198 | 64 | 0.0254 |
| PAGEBLOCKS | 5393 | 10 | 0.0946 |
| PENDIGITS | 6870 | 16 | 0.0227 |
| PIMA | 768 | 8 | 0.3490 |
| SATELLITE | 6435 | 36 | 0.3164 |
| SATIMAGE | 5801 | 36 | 0.0119 |
| SHUTTLE | 20000 | 9 | 0.0725 |
| THYROID | 3656 | 6 | 0.0254 |
| VERTEBRAL | 240 | 6 | 0.1250 |
| VOWELS | 1452 | 12 | 0.0317 |
| WAVEFORM | 3443 | 21 | 0.0290 |
| WBC | 223 | 9 | 0.0448 |
| WDBC | 367 | 30 | 0.0272 |
| WILT | 4819 | 5 | 0.0533 |
| WINE | 129 | 13 | 0.0775 |
| WPBC | 198 | 33 | 0.2374 |
| YEAST | 1453 | 8 | 0.3310 |

Table 4: Cost per example (mean $\pm$ std) per detector aggregated over the datasets. Results show that REJEX obtains a lower average cost for 9 out of 12 detectors and similar average cost as the runner-up SS-REPEN for the remaining 3 detectors. Moreover, REJEX has the best overall average (last row).

| Det. | COST PER EXAMPLE (MEAN $\pm$ STD.) | | | | | | | |
|---|---|---|---|---|---|---|---|---|
| | REJEX | SS-REPEN | MV | ENS | UDR | EM | STABILITY | NOREJECT |
| AE | **0.122 ± 0.139** | 0.124 ± 0.133 | 0.136 ± 0.143 | 0.134 ± 0.151 | 0.138 ± 0.148 | 0.143 ± 0.150 | 0.143 ± 0.149 | 0.148 ± 0.152 |
| COPOD | **0.123 ± 0.138** | 0.125 ± 0.134 | 0.135 ± 0.142 | 0.133 ± 0.142 | 0.140 ± 0.144 | 0.143 ± 0.148 | 0.145 ± 0.147 | 0.146 ± 0.148 |
| ECOD | **0.120 ± 0.136** | 0.124 ± 0.135 | 0.129 ± 0.136 | 0.133 ± 0.143 | 0.139 ± 0.142 | 0.140 ± 0.145 | 0.143 ± 0.144 | 0.145 ± 0.145 |
| GMM | **0.122 ± 0.135** | 0.124 ± 0.137 | 0.144 ± 0.141 | 0.136 ± 0.146 | 0.142 ± 0.145 | 0.156 ± 0.148 | 0.151 ± 0.147 | 0.156 ± 0.149 |
| HBOS | **0.116 ± 0.129** | 0.123 ± 0.138 | 0.131 ± 0.132 | 0.134 ± 0.136 | 0.135 ± 0.137 | 0.139 ± 0.141 | 0.140 ± 0.139 | 0.142 ± 0.142 |
| IFOR | **0.115 ± 0.128** | 0.123 ± 0.135 | 0.130 ± 0.136 | 0.129 ± 0.136 | 0.134 ± 0.139 | 0.140 ± 0.143 | 0.139 ± 0.141 | 0.143 ± 0.144 |
| INNE | **0.113 ± 0.129** | 0.122 ± 0.133 | 0.145 ± 0.134 | 0.146 ± 0.140 | 0.145 ± 0.138 | 0.147 ± 0.140 | 0.146 ± 0.139 | 0.145 ± 0.140 |
| KDE | **0.127 ± 0.140** | **0.127 ± 0.134** | 0.143 ± 0.145 | 0.138 ± 0.145 | 0.144 ± 0.145 | 0.150 ± 0.148 | 0.145 ± 0.143 | 0.152 ± 0.148 |
| KNN | **0.119 ± 0.123** | 0.125 ± 0.135 | 0.140 ± 0.131 | 0.135 ± 0.131 | 0.135 ± 0.130 | 0.144 ± 0.132 | 0.141 ± 0.131 | 0.146 ± 0.133 |
| LODA | **0.125 ± 0.133** | 0.125 ± 0.134 | 0.131 ± 0.130 | 0.137 ± 0.137 | 0.140 ± 0.136 | 0.146 ± 0.141 | 0.141 ± 0.131 | 0.151 ± 0.142 |
| LOF | **0.126 ± 0.131** | **0.126 ± 0.136** | 0.155 ± 0.140 | 0.140 ± 0.139 | 0.142 ± 0.138 | 0.157 ± 0.140 | 0.151 ± 0.139 | 0.158 ± 0.140 |
| OCSVM | **0.120 ± 0.131** | 0.125 ± 0.133 | 0.138 ± 0.138 | 0.132 ± 0.140 | 0.138 ± 0.140 | 0.141 ± 0.140 | 0.137 ± 0.136 | 0.147 ± 0.143 |
| AVG. | **0.121 ± 0.133** | 0.125 ± 0.135 | 0.138 ± 0.137 | 0.136 ± 0.140 | 0.139 ± 0.140 | 0.146 ± 0.143 | 0.144 ± 0.140 | 0.148 ± 0.144 |

Table 5: Ranking positions (mean ± std) per detector aggregated over the datasets. Results show that REJEX obtains always the lowest average rank, despite being close to the runner-up SS-REPEN when the detector is LODA.

| DET. | | RANKING POSITION (MEAN ± STD.) | | | | | | |
|---|---|---|---|---|---|---|---|---|
| | REJEX | SS-REPEN | MV | ENS | UDR | EM | STABILITY | NOREJECT |
| AE | **2.63 ± 1.63** | 3.96 ± 2.94 | 4.78 ± 2.10 | 3.81 ± 2.11 | 4.98 ± 1.88 | 5.15 ± 1.80 | 4.92 ± 1.78 | 5.77 ± 1.84 |
| COPOD | **2.49 ± 1.65** | 3.44 ± 2.75 | 3.97 ± 1.92 | 4.25 ± 2.17 | 5.24 ± 1.70 | 5.13 ± 1.67 | 5.59 ± 1.48 | 5.89 ± 2.13 |
| ECOD | **2.43 ± 1.44** | 3.62 ± 2.86 | 3.73 ± 1.96 | 4.13 ± 2.29 | 5.53 ± 1.57 | 4.75 ± 1.62 | 5.74 ± 1.39 | 6.07 ± 1.93 |
| GMM | **2.12 ± 1.08** | 3.05 ± 2.49 | 5.26 ± 1.92 | 3.49 ± 2.00 | 4.43 ± 1.66 | 6.26 ± 1.24 | 5.20 ± 1.43 | 6.20 ± 1.45 |
| HBOS | **2.29 ± 1.57** | 3.64 ± 2.98 | 4.54 ± 2.11 | 4.39 ± 2.06 | 4.95 ± 1.79 | 5.04 ± 1.80 | 5.61 ± 1.40 | 5.52 ± 1.88 |
| IFOR | **2.23 ± 1.48** | 3.78 ± 2.78 | 4.12 ± 1.90 | 4.26 ± 2.08 | 5.10 ± 1.88 | 5.27 ± 1.66 | 5.34 ± 1.38 | 5.91 ± 2.22 |
| INNE | **1.73 ± 1.14** | 3.18 ± 2.74 | 5.86 ± 2.42 | 5.57 ± 1.40 | 4.94 ± 1.62 | 5.57 ± 1.60 | 5.32 ± 1.37 | 3.83 ± 1.63 |
| KDE | **2.33 ± 1.42** | 3.99 ± 2.86 | 4.74 ± 2.06 | 3.79 ± 2.03 | 5.01 ± 1.90 | 5.43 ± 1.59 | 4.87 ± 1.92 | 5.84 ± 1.80 |
| KNN | **2.02 ± 1.29** | 3.58 ± 2.87 | 4.87 ± 1.81 | 3.94 ± 1.94 | 4.41 ± 1.83 | 5.75 ± 1.49 | 5.22 ± 1.62 | 6.21 ± 1.48 |
| LODA | **2.89 ± 1.77** | 3.17 ± 2.30 | 4.15 ± 2.26 | 4.36 ± 2.14 | 4.99 ± 2.00 | 5.50 ± 2.04 | 4.98 ± 2.11 | 5.95 ± 1.73 |
| LOF | **2.04 ± 1.01** | 3.16 ± 2.73 | 5.68 ± 1.40 | 3.32 ± 1.71 | 3.96 ± 1.63 | 6.15 ± 1.19 | 5.47 ± 1.49 | 6.22 ± 1.31 |
| OCSVM | **2.33 ± 1.29** | 3.92 ± 2.84 | 4.89 ± 1.98 | 3.85 ± 2.17 | 4.86 ± 1.89 | 5.31 ± 1.80 | 5.06 ± 1.89 | 5.78 ± 1.66 |
| AVG. | **2.29 ± 1.40** | 3.54 ± 2.76 | 4.72 ± 1.99 | 4.10 ± 2.01 | 4.87 ± 1.78 | 5.44 ± 1.63 | 5.28 ± 1.60 | 5.77 ± 1.76 |

Table 6: Cost per example (mean ± std) per detector aggregated over the datasets. The table is divided into three parts, where each part has different costs (false positives, false negatives, rejection). Results show that REJEX obtains a lower average cost in all cases but one (KDE).

| DET. | | COST PER EXAMPLE FOR THREE COST FUNCTIONS (MEAN ± STD) | | | | | | |
|---|---|---|---|---|---|---|---|---|
| | REJEX | SS-REPEN | MV | ENS | UDR | EM | STABILITY | NOREJECT |
| | FALSE POSITIVE COST = 10, FALSE NEGATIVE COST = 1, REJECTION COST = $\min\{10(1-\gamma), \gamma\}$ | | | | | | | |
| AE | **0.504 ± 0.626** | 0.584 ± 0.723 | 0.697 ± 0.763 | 0.661 ± 0.830 | 0.703 ± 0.829 | 0.766 ± 0.841 | 0.768 ± 0.826 | 0.825 ± 0.873 |
| COPOD | **0.491 ± 0.637** | 0.593 ± 0.706 | 0.686 ± 0.746 | 0.618 ± 0.726 | 0.707 ± 0.788 | 0.778 ± 0.825 | 0.785 ± 0.801 | 0.781 ± 0.833 |
| ECOD | **0.479 ± 0.628** | 0.584 ± 0.727 | 0.625 ± 0.705 | 0.642 ± 0.755 | 0.711 ± 0.774 | 0.748 ± 0.803 | 0.770 ± 0.783 | 0.771 ± 0.817 |
| GMM | **0.568 ± 0.713** | 0.589 ± 0.752 | 0.823 ± 0.878 | 0.715 ± 0.929 | 0.790 ± 0.925 | 0.941 ± 0.948 | 0.889 ± 0.929 | 0.950 ± 0.967 |
| HBOS | **0.475 ± 0.595** | 0.569 ± 0.758 | 0.666 ± 0.693 | 0.697 ± 0.732 | 0.709 ± 0.764 | 0.776 ± 0.803 | 0.771 ± 0.770 | 0.809 ± 0.816 |
| IFOR | **0.477 ± 0.602** | 0.575 ± 0.712 | 0.665 ± 0.718 | 0.634 ± 0.731 | 0.683 ± 0.786 | 0.776 ± 0.818 | 0.763 ± 0.788 | 0.808 ± 0.831 |
| INNE | **0.479 ± 0.592** | 0.567 ± 0.698 | 0.752 ± 0.724 | 0.820 ± 0.795 | 0.815 ± 0.787 | 0.819 ± 0.793 | 0.818 ± 0.792 | 0.823 ± 0.799 |
| KDE | 0.602 ± 0.827 | **0.589 ± 0.704** | 0.819 ± 0.947 | 0.740 ± 0.913 | 0.793 ± 0.939 | 0.897 ± 0.945 | 0.774 ± 0.906 | 0.914 ± 0.945 |
| KNN | **0.498 ± 0.577** | 0.596 ± 0.726 | 0.741 ± 0.734 | 0.669 ± 0.720 | 0.669 ± 0.736 | 0.777 ± 0.747 | 0.739 ± 0.735 | 0.800 ± 0.749 |
| LODA | **0.518 ± 0.619** | 0.574 ± 0.709 | 0.574 ± 0.647 | 0.689 ± 0.729 | 0.701 ± 0.748 | 0.762 ± 0.774 | 0.697 ± 0.682 | 0.827 ± 0.797 |
| LOF | **0.539 ± 0.623** | 0.603 ± 0.742 | 0.898 ± 0.840 | 0.685 ± 0.773 | 0.715 ± 0.790 | 0.891 ± 0.813 | 0.831 ± 0.821 | 0.887 ± 0.808 |
| OCSVM | **0.479 ± 0.599** | 0.589 ± 0.705 | 0.745 ± 0.790 | 0.632 ± 0.752 | 0.694 ± 0.782 | 0.760 ± 0.775 | 0.695 ± 0.737 | 0.818 ± 0.806 |
| | FALSE POSITIVE COST = 1, FALSE NEGATIVE COST = 10, REJECTION COST = $\min\{1-\gamma, 10\gamma\}$ | | | | | | | |
| AE | **0.730 ± 0.747** | 0.761 ± 0.756 | 0.909 ± 0.882 | 0.784 ± 0.825 | 0.780 ± 0.805 | 0.819 ± 0.843 | 0.789 ± 0.825 | 0.797 ± 0.821 |
| COPOD | **0.761 ± 0.767** | 0.765 ± 0.770 | 0.930 ± 0.888 | 0.794 ± 0.805 | 0.800 ± 0.801 | 0.844 ± 0.842 | 0.802 ± 0.815 | 0.827 ± 0.832 |
| ECOD | **0.739 ± 0.759** | 0.767 ± 0.766 | 0.900 ± 0.858 | 0.789 ± 0.811 | 0.788 ± 0.787 | 0.840 ± 0.839 | 0.791 ± 0.803 | 0.821 ± 0.819 |
| GMM | **0.670 ± 0.676** | 0.765 ± 0.767 | 0.845 ± 0.782 | 0.754 ± 0.755 | 0.739 ± 0.736 | 0.785 ± 0.757 | 0.760 ± 0.753 | 0.766 ± 0.750 |
| HBOS | **0.687 ± 0.684** | 0.776 ± 0.782 | 0.824 ± 0.808 | 0.750 ± 0.768 | 0.744 ± 0.747 | 0.785 ± 0.787 | 0.749 ± 0.765 | 0.753 ± 0.766 |
| IFOR | **0.679 ± 0.680** | 0.775 ± 0.776 | 0.847 ± 0.824 | 0.755 ± 0.771 | 0.743 ± 0.745 | 0.761 ± 0.772 | 0.757 ± 0.774 | 0.763 ± 0.770 |
| INNE | **0.660 ± 0.685** | 0.772 ± 0.779 | 0.695 ± 0.620 | 0.774 ± 0.742 | 0.748 ± 0.722 | 0.758 ± 0.737 | 0.744 ± 0.716 | 0.773 ± 0.754 |
| KDE | **0.691 ± 0.692** | 0.791 ± 0.773 | 0.887 ± 0.836 | 0.754 ± 0.760 | 0.755 ± 0.744 | 0.785 ± 0.807 | 0.758 ± 0.754 | 0.759 ± 0.760 |
| KNN | **0.706 ± 0.657** | 0.767 ± 0.764 | 0.839 ± 0.779 | 0.791 ± 0.736 | 0.769 ± 0.710 | 0.778 ± 0.736 | 0.799 ± 0.736 | 0.803 ± 0.729 |
| LODA | **0.750 ± 0.714** | 0.781 ± 0.775 | 0.880 ± 0.850 | 0.811 ± 0.768 | 0.806 ± 0.761 | 0.804 ± 0.783 | 0.827 ± 0.780 | 0.838 ± 0.784 |
| LOF | **0.738 ± 0.679** | 0.764 ± 0.764 | 0.999 ± 0.833 | 0.826 ± 0.757 | 0.810 ± 0.739 | 0.871 ± 0.770 | 0.867 ± 0.799 | 0.846 ± 0.747 |
| OCSVM | **0.730 ± 0.711** | 0.780 ± 0.774 | 0.953 ± 0.878 | 0.791 ± 0.786 | 0.795 ± 0.773 | 0.845 ± 0.833 | 0.787 ± 0.772 | 0.796 ± 0.783 |
| | FALSE POSITIVE COST = 5, FALSE NEGATIVE COST = 5, REJECTION COST = $\gamma$ | | | | | | | |
| AE | **0.534 ± 0.611** | 0.618 ± 0.666 | 0.671 ± 0.716 | 0.644 ± 0.741 | 0.655 ± 0.736 | 0.707 ± 0.748 | 0.705 ± 0.740 | 0.738 ± 0.762 |
| COPOD | **0.545 ± 0.619** | 0.627 ± 0.673 | 0.676 ± 0.724 | 0.629 ± 0.674 | 0.666 ± 0.716 | 0.719 ± 0.747 | 0.719 ± 0.730 | 0.731 ± 0.739 |
| ECOD | **0.529 ± 0.609** | 0.625 ± 0.675 | 0.629 ± 0.687 | 0.638 ± 0.702 | 0.662 ± 0.705 | 0.701 ± 0.736 | 0.708 ± 0.716 | 0.724 ± 0.727 |
| GMM | **0.534 ± 0.599** | 0.626 ± 0.687 | 0.719 ± 0.709 | 0.656 ± 0.716 | 0.675 ± 0.720 | 0.776 ± 0.736 | 0.746 ± 0.731 | 0.780 ± 0.743 |
| HBOS | **0.499 ± 0.572** | 0.622 ± 0.694 | 0.632 ± 0.661 | 0.650 ± 0.669 | 0.641 ± 0.681 | 0.695 ± 0.706 | 0.688 ± 0.686 | 0.710 ± 0.709 |
| IFOR | **0.497 ± 0.569** | 0.623 ± 0.677 | 0.643 ± 0.680 | 0.620 ± 0.667 | 0.629 ± 0.692 | 0.696 ± 0.712 | 0.688 ± 0.701 | 0.714 ± 0.719 |
| INNE | **0.491 ± 0.569** | 0.617 ± 0.668 | 0.622 ± 0.572 | 0.718 ± 0.691 | 0.691 ± 0.674 | 0.709 ± 0.685 | 0.705 ± 0.675 | 0.726 ± 0.698 |
| KDE | **0.562 ± 0.639** | 0.642 ± 0.673 | 0.709 ± 0.739 | 0.666 ± 0.711 | 0.684 ± 0.726 | 0.748 ± 0.742 | 0.679 ± 0.689 | 0.761 ± 0.742 |
| KNN | **0.521 ± 0.544** | 0.628 ± 0.677 | 0.687 ± 0.667 | 0.657 ± 0.646 | 0.634 ± 0.651 | 0.701 ± 0.664 | 0.693 ± 0.657 | 0.728 ± 0.664 |
| LODA | **0.550 ± 0.595** | 0.627 ± 0.677 | 0.601 ± 0.649 | 0.670 ± 0.668 | 0.665 ± 0.680 | 0.708 ± 0.698 | 0.683 ± 0.645 | 0.757 ± 0.711 |
| LOF | **0.554 ± 0.580** | 0.628 ± 0.681 | 0.809 ± 0.737 | 0.678 ± 0.682 | 0.674 ± 0.688 | 0.792 ± 0.703 | 0.759 ± 0.718 | 0.788 ± 0.698 |
| OCSVM | **0.523 ± 0.582** | 0.631 ± 0.671 | 0.704 ± 0.728 | 0.634 ± 0.685 | 0.657 ± 0.695 | 0.709 ± 0.704 | 0.660 ± 0.674 | 0.733 ± 0.716 |

Table 7: Rankings (mean ± std) per detector aggregated over the datasets, where lower positions mean lower costs (better). The table is divided into three parts, where each part has different costs for false positives, false negatives, and rejection. REJEX obtains the lowest (best) average ranking position for all the detectors and all cost functions.

| Det. | RANKINGS FOR THE THREE COST FUNCTIONS (MEAN ± STD) | | | | | | | |
|---|---|---|---|---|---|---|---|---|
| | REJEX | SS-REPEN | MV | ENS | UDR | EM | STABILITY | NOREJECT |
| **FALSE POSITIVE COST = 10, FALSE NEGATIVE COST = 1, REJECTION COST = $\min\{10(1-\gamma), \gamma\}$** | | | | | | | | |
| AE | **2.35 ± 1.37** | 3.84 ± 2.73 | 5.87 ± 2.40 | 3.68 ± 2.05 | 4.85 ± 1.96 | 5.33 ± 1.67 | 4.72 ± 1.68 | 5.36 ± 1.97 |
| COPOD | **2.25 ± 1.45** | 3.63 ± 2.66 | 4.79 ± 2.27 | 3.89 ± 2.27 | 4.94 ± 1.76 | 5.46 ± 1.71 | 5.51 ± 1.45 | 5.54 ± 2.17 |
| ECOD | **2.28 ± 1.30** | 3.51 ± 2.73 | 4.63 ± 2.36 | 3.85 ± 2.21 | 5.34 ± 1.74 | 5.11 ± 1.66 | 5.38 ± 1.39 | 5.90 ± 2.09 |
| GMM | **2.13 ± 0.99** | 3.10 ± 2.43 | 6.36 ± 2.23 | 3.31 ± 1.94 | 4.21 ± 1.69 | 6.28 ± 1.27 | 5.18 ± 1.53 | 5.44 ± 1.50 |
| HBOS | **2.12 ± 1.42** | 3.46 ± 2.85 | 5.41 ± 2.42 | 4.25 ± 2.06 | 4.78 ± 1.78 | 5.35 ± 1.66 | 5.42 ± 1.47 | 5.20 ± 1.95 |
| IFOR | **2.11 ± 1.49** | 3.69 ± 2.61 | 4.73 ± 2.26 | 4.02 ± 2.11 | 5.07 ± 1.87 | 5.39 ± 1.61 | 5.36 ± 1.41 | 5.63 ± 2.24 |
| INNE | **1.72 ± 1.24** | 3.09 ± 2.68 | 5.42 ± 2.45 | 6.16 ± 1.44 | 5.47 ± 1.59 | 4.60 ± 1.51 | 5.32 ± 1.31 | 4.21 ± 1.80 |
| KDE | **2.14 ± 1.25** | 3.82 ± 2.68 | 5.73 ± 2.36 | 3.54 ± 1.92 | 4.75 ± 1.85 | 5.84 ± 1.58 | 4.83 ± 1.91 | 5.36 ± 1.75 |
| KNN | **1.99 ± 1.28** | 3.50 ± 2.74 | 5.55 ± 2.21 | 3.92 ± 2.02 | 4.37 ± 1.86 | 5.73 ± 1.46 | 5.23 ± 1.68 | 5.71 ± 1.71 |
| LODA | **2.56 ± 1.53** | 3.31 ± 2.31 | 4.29 ± 2.48 | 4.34 ± 2.14 | 5.03 ± 1.95 | 5.42 ± 1.88 | 4.96 ± 2.04 | 6.08 ± 1.75 |
| LOF | **1.96 ± 1.03** | 3.04 ± 2.46 | 7.12 ± 1.28 | 3.14 ± 1.60 | 3.73 ± 1.31 | 6.27 ± 1.14 | 5.59 ± 1.66 | 5.15 ± 1.39 |
| OCSVM | **2.15 ± 1.20** | 3.93 ± 2.70 | 5.92 ± 2.30 | 3.58 ± 2.13 | 4.70 ± 1.92 | 5.40 ± 1.63 | 5.03 ± 1.89 | 5.29 ± 1.67 |
| **FALSE POSITIVE COST = 1, FALSE NEGATIVE COST = 10, REJECTION COST = $\min\{1-\gamma, 10\gamma\}$** | | | | | | | | |
| AE | **2.98 ± 1.93** | 3.82 ± 2.72 | 7.03 ± 1.95 | 4.30 ± 2.07 | 4.49 ± 1.82 | 4.96 ± 1.83 | 4.28 ± 1.62 | 4.14 ± 1.93 |
| COPOD | **2.91 ± 2.04** | 3.56 ± 2.69 | 7.13 ± 1.56 | 4.15 ± 1.97 | 4.43 ± 1.87 | 5.30 ± 1.86 | 4.50 ± 1.60 | 4.03 ± 1.79 |
| ECOD | **2.70 ± 1.96** | 3.88 ± 2.82 | 6.87 ± 1.72 | 4.15 ± 2.02 | 4.74 ± 1.79 | 5.01 ± 2.03 | 4.23 ± 1.50 | 4.42 ± 1.90 |
| GMM | **2.59 ± 1.70** | 3.99 ± 2.85 | 6.84 ± 2.22 | 4.04 ± 2.12 | 4.08 ± 1.77 | 5.73 ± 1.37 | 4.62 ± 1.55 | 4.12 ± 1.58 |
| HBOS | **2.96 ± 2.14** | 4.32 ± 2.93 | 6.41 ± 2.20 | 4.49 ± 1.92 | 4.37 ± 1.82 | 5.15 ± 1.94 | 4.48 ± 1.65 | 3.81 ± 1.81 |
| IFOR | **2.71 ± 2.06** | 4.51 ± 2.92 | 6.80 ± 2.00 | 4.47 ± 2.09 | 4.62 ± 1.72 | 4.33 ± 1.66 | 4.55 ± 1.47 | 4.01 ± 1.93 |
| INNE | **2.64 ± 1.94** | 4.71 ± 2.93 | 5.06 ± 2.95 | 5.85 ± 1.52 | 5.06 ± 1.80 | 4.03 ± 1.64 | 4.72 ± 1.50 | 3.94 ± 1.87 |
| KDE | **3.00 ± 2.01** | 4.49 ± 2.93 | 6.51 ± 2.27 | 4.00 ± 1.84 | 4.40 ± 1.68 | 5.01 ± 1.97 | 4.40 ± 1.78 | 4.18 ± 1.96 |
| KNN | **2.64 ± 2.01** | 4.11 ± 3.01 | 6.67 ± 2.23 | 4.17 ± 1.89 | 4.13 ± 1.87 | 4.99 ± 1.60 | 4.88 ± 1.54 | 4.41 ± 1.64 |
| LODA | **3.44 ± 1.96** | 3.66 ± 2.71 | 6.32 ± 2.30 | 4.22 ± 1.94 | 4.36 ± 1.95 | 4.47 ± 2.17 | 4.53 ± 2.09 | 4.99 ± 1.87 |
| LOF | **2.22 ± 1.38** | 3.43 ± 2.67 | 7.74 ± 0.67 | 3.47 ± 1.73 | 3.63 ± 1.40 | 5.95 ± 1.17 | 5.35 ± 1.57 | 4.22 ± 1.38 |
| OCSVM | **2.82 ± 1.71** | 3.83 ± 2.63 | 7.30 ± 1.50 | 4.23 ± 2.13 | 4.35 ± 1.78 | 5.34 ± 1.65 | 4.32 ± 1.95 | 3.80 ± 1.72 |
| **FALSE POSITIVE COST = 5, FALSE NEGATIVE COST = 5, REJECTION COST = $\gamma$** | | | | | | | | |
| AE | **2.31 ± 1.38** | 4.05 ± 2.78 | 5.85 ± 2.41 | 3.66 ± 2.12 | 4.69 ± 1.86 | 5.27 ± 1.68 | 4.84 ± 1.74 | 5.34 ± 1.92 |
| COPOD | **2.24 ± 1.49** | 3.72 ± 2.70 | 4.62 ± 2.17 | 3.98 ± 2.34 | 4.89 ± 1.87 | 5.26 ± 1.58 | 5.57 ± 1.50 | 5.72 ± 2.10 |
| ECOD | **2.18 ± 1.31** | 3.92 ± 2.75 | 4.22 ± 2.22 | 3.93 ± 2.33 | 5.30 ± 1.82 | 4.91 ± 1.69 | 5.48 ± 1.38 | 6.06 ± 1.94 |
| GMM | **1.96 ± 0.97** | 3.31 ± 2.44 | 6.39 ± 2.21 | 3.36 ± 1.89 | 4.09 ± 1.68 | 6.29 ± 1.25 | 5.11 ± 1.51 | 5.48 ± 1.50 |
| HBOS | **1.98 ± 1.37** | 3.95 ± 2.88 | 5.30 ± 2.46 | 4.18 ± 2.01 | 4.63 ± 1.83 | 5.36 ± 1.68 | 5.41 ± 1.46 | 5.18 ± 1.93 |
| IFOR | **2.01 ± 1.46** | 4.10 ± 2.67 | 4.71 ± 2.26 | 3.96 ± 2.05 | 4.98 ± 1.96 | 5.35 ± 1.60 | 5.29 ± 1.42 | 5.60 ± 2.24 |
| INNE | **1.70 ± 1.25** | 3.75 ± 2.82 | 4.17 ± 2.42 | 6.02 ± 1.44 | 5.57 ± 1.78 | 4.56 ± 1.36 | 5.36 ± 1.38 | 4.87 ± 2.08 |
| KDE | **2.22 ± 1.35** | 4.24 ± 2.73 | 5.49 ± 2.55 | 3.62 ± 2.00 | 4.71 ± 1.95 | 5.60 ± 1.50 | 4.79 ± 1.86 | 5.34 ± 1.87 |
| KNN | **1.98 ± 1.23** | 3.88 ± 2.82 | 5.49 ± 2.39 | 3.91 ± 1.86 | 4.29 ± 1.86 | 5.56 ± 1.71 | 5.19 ± 1.63 | 5.69 ± 1.70 |
| LODA | **2.58 ± 1.60** | 3.59 ± 2.36 | 4.34 ± 2.58 | 4.26 ± 2.17 | 4.93 ± 1.98 | 5.33 ± 1.98 | 5.02 ± 1.98 | 5.94 ± 1.75 |
| LOF | **1.88 ± 0.96** | 3.26 ± 2.51 | 7.18 ± 1.29 | 3.16 ± 1.60 | 3.65 ± 1.32 | 6.24 ± 1.12 | 5.53 ± 1.69 | 5.10 ± 1.37 |
| OCSVM | **2.15 ± 1.19** | 4.18 ± 2.77 | 5.73 ± 2.36 | 3.62 ± 2.20 | 4.59 ± 1.88 | 5.39 ± 1.59 | 5.05 ± 1.92 | 5.31 ± 1.67 |

