# OpenReview forum: "Unsupervised Anomaly Detection with Rejection"
_NeurIPS.cc/2023/Conference — NeurIPS 2023 poster_

### Official Review · Reviewer_imnX · 2023-07-05

**Soundness:** 3 good
**Presentation:** 2 fair
**Contribution:** 3 good
**Rating:** 6
**Confidence:** 2

**Summary:**

The authors address the topic of rejection of samples in an unsupervised anomaly detection setup. Their approach focuses on determining a constant rejection threshold, which allows the detector to reject examples with high uncertainty. The new proposed method introduces this rejection threshold based on a confidence score given by another existing model (ExCeed). The authors provide theoretical analyses as well as empirical experiments and show that it is possible to set a constant rejection threshold with strong theoretical guarantees.


**Strengths:**

- The paper provides a good structure but also requires some pre-knowledge in this area to be able to follow the presented thoughts e. g. research questions are later connected according to sections.
- In-depth theoretical methodology as well as empirical evaluation.
- Detailed overview of single results in the supplement is given.

**Weaknesses:**

- It was not always straightforward to follow the paper, especially because a lot of variables are introduced but are defined much later in the paper (e.g. t_1(n, \gamma, T) (line 131) and  \gamma = \mathds{P}(Y = 1) (line 199)). Starting with a more explanatory part would let the reader build an intuition about which factors are important to calculate the rejection threshold. With all the factors in mind, it will get easier to follow the complex theoretical contribution.
- in line 135 \epsilon is defined as 1 - 2e^(-T). In the formula between lines 123 and 124, the rejection threshold is defined as \mathcal{T} = 1 - \epsilon = 1 - 2e^(-T), which would result in \epsilon = 2e^(-T).

Minor comments:
- "Our approach is called **RejEx** (Rejecting via ExCeed)(line 110)
- In Theorem 3.8 it is refed to Theorem 3.5 for the definition of g. Theorem 3.5, however, refers to Theorem 3.4; It would be easier to directly refer to Theorem 3.4


**Questions:**

None

**Limitations:**

Difficult write-up.

---

> ### Author Rebuttal · Authors · 2023-08-08
>
> Dear Reviewer imnX,
>
> We appreciate your **positive feedback**, and that you eventually were able to **follow our theoretical contribution**. In the revised version, we will address your points about the notation and try to give some **more intuitions prior to the theoretical sections**.
>
> We will fix the typos you correctly pointed out: line 135 should have $2e^{-T}$ instead of $1-2e^{-T}$, we missed *RejEx* in line 110, and Theorem 3.8 should refer to Def. 3.4 instead of Theorem 3.5.

---

> > ### Comment · Reviewer_imnX · 2023-08-16
> > **rebuttal**
> >
> > thx.

---

### Official Review · Reviewer_Xnvu · 2023-07-05

**Soundness:** 3 good
**Presentation:** 3 good
**Contribution:** 3 good
**Rating:** 6
**Confidence:** 4

**Summary:**

This paper presents a rejection scheme for the task of unsupervised anomaly detection.  Learning to reject enables a predictor to withhold from making a prediction; this paradigm is more common in unsupervised learning. Here, the authors extend the rejection idea to the unsupervised anomaly detection task.  The idea is to reject samples based on a stability metric; namely, of the prediction is unstable to small changes in the feature space, the prediction is rejected. This type of stability metric was recently proposed and is termed EXCEED. The authors present a theoretical analysis of the EXCEED metric and derive upper bounds for the test rejection rate and expected prediction cost. The new scheme is evaluated for several anomaly detectors on real datasets and outperforms other rejection schemes.

**Strengths:**

The paper is well-written, and easy to follow.  Overall, the presentation is scientifically sound. The problem of unsupervised anomaly detection is extremely challenging and important; the paper presents a rejection scheme that could improve trust in commonly used detectors. The idea of using stability and specifically the EXCEED metric, makes sense. The theoretical analysis strengthens the work and offers bounds on the expected values of the presented scheme. The empirical evidence presented in the paper is promising and demonstrates the merits of the method.

**Weaknesses:**

Background on the EXCEED method is missing; adding more information on this metric could help the reader. Some recently proposed NN anomaly detectors are missing from the evaluation, for example:
[1] Qiu, Chen, et al. "Neural transformation learning for deep anomaly detection beyond images." International Conference on Machine Learning. PMLR, 2021.
[2] Shenkar, T., & Wolf, L. (2021, October). Anomaly detection for tabular data with internal contrastive learning. In International Conference on Learning Representations.
[3] Lindenbaum, et al. (2021). Probabilistic robust autoencoders for outlier detection. arXiv preprint arXiv:2110.00494.
There are many other NN that could be included, I think some NN baselines should be considered.
The description of the experiments conducted is too brief; it would be good if the authors could expand on the implementation and evaluation protocol. For example, what is $\lambda$ in all experiments? Or how is it tuned?

Sample size is limited in the evaluation to 20K.

**Questions:**

The analysis assumes that $\gamma$ is known, in practice how can you gain access to this value? Is there a way to overcome this limitation?
Please expand on the ECEED metric, equation 1.
A comma is missing after this equation.
How is the cost influenced when changing \lambda? It is not clear from the text what do you do with this value.
Why are you not mentioning the subsampling in the main text? Is it so computationally demanding to evaluate the method on large datasets?


**Limitations:**

Limitations are discussed in the last section, I would however add information on the cases in which rejection increases the cost, for example using statistic across datasets.

---

> ### Author Rebuttal · Authors · 2023-08-08
>
> Dear Reviewer Xnvu,
>
> Thanks for your **positive and constructive feedback**. Here are our responses:
>
> 1.  [**Experimental setup**] Section 5.1 clarifies our experimental setup, including **how we set hyperparameters**. If you can let us know specific things that are unclear, we will add them. Also, we have submitted and will release all information (code, data repository, …) to replicate the experiments.
>
> 2.  [**Setting the decision threshold $\lambda$**] As stated in lines 71-73, the decision threshold $\lambda$ is set such that $\gamma \times n$ scores are $\ge \lambda$, where $\gamma$ is the dataset’s contamination factor and $n$ is the training set size. Because we are operating in a **fully unsupervised** setting, we do not consider $\lambda$ as a hyperparameter to tune [59] but we set it the same way as in other papers [see for example cites 23,49,50,51]. Consequently, we do not analyze how the test cost varies when changing $\lambda$.
>
> 3.  [**$\gamma$ is known**] We assume that the contamination factor $\gamma$ is given, as stated in line 98. However, approaches exist to **estimate it** from a given dataset. This can even be done from a **fully unlabeled** dataset; see for example cite 50 in the paper.
>
> 4.  [**References**] We will discuss the references in the final version of the paper and will do our best to **include as many as possible** in the experimental analysis.
>
> 5.  [**ExCeeD**] We will provide further details about exceed in the final version of the paper. Also, see our response to reviewer 3a9y.
>
> 6.  [**Computational cost**] Limiting the dataset size to $20K$ by taking a subsample is an experimental detail that has the unique goal of **saving computational effort**. In fact, running all $2040$ experiments with the size limit requires more than a week. Note that, as shown in Q3, our method has **low computational cost**, as opposed to *Stability*, which uses an expensive internal optimization, and *Ens*, which uses an ensemble of models.

---

> > ### Comment · Reviewer_Xnvu · 2023-08-18
> > **Response to authors**
> >
> > I thank the authors for responding to all my comments.
> > I have no additional open questions about the paper.

---

### Official Review · Reviewer_R9xH · 2023-07-06

**Soundness:** 4 excellent
**Presentation:** 4 excellent
**Contribution:** 4 excellent
**Rating:** 5
**Confidence:** 3

**Summary:**

This paper suggests applying the stability metric computed by EXCEED for anomaly detection. The authors present theoretical findings regarding this metric, including the test rejection rate, as well as upper bounds for both the rejection rate and the expected prediction cost. Furthermore, comprehensive experiments are conducted to validate the effectiveness of the proposed method.

**Strengths:**

1. The presentation of the paper is clear, and the proposed method is simple but effective.
2. This paper offers a theoretical analysis of EXCEED, deriving the upper bounds for both the rejection rate and the expected prediction cost.
3. The effectiveness of the proposed method and the validity of the theoretical results are confirmed through comprehensive experiments.

**Weaknesses:**

1. The methods compared in Figure 1 appear to be significantly dated. It would be valuable if the paper could include additional results pertaining to recently proposed methods.

**Questions:**

Please refer to [Weakness].

**Limitations:**

Please refer to [Weakness].

---

> ### Author Rebuttal · Authors · 2023-08-08
>
> Dear Reviewer R9xH,
>
> We appreciate your **positive review**. Our method is **anomaly detector-agnostic**, which means that it can be applied on top of any anomaly detector. We ran the experiments using $12$ anomaly detectors included in the most recent and largest experimental comparison [23], which include:
>
> -  Classical algorithms, like LOF and IForest, that are often used as baselines because they obtain competitive performance [65, 66, 67];
>
> -  More recent algorithms, like COPOD (2020), ECOD (2022).
>
> Note that cite [23] claims that **“none of the unsupervised methods is statistically better than the others”** and that **“some Deep Learning based unsupervised methods are surprisingly worse than shallow methods”**.
>
> Furthermore, out of $34$ datasets, $15$ datasets have at least one detector with $AUC > 0.90$, and $12$ datasets have at least one detector with $AUC > 0.7$. Thus this seems like a **reasonably extensive benchmark**.
>
> Finally, from the review, it is unclear what other anomaly detection methods the reviewer considers to be state-of-the-art.
>
> ----
>
> [65] Qiu Chen, et al. "Neural transformation learning for deep anomaly detection beyond images." International Conference on Machine Learning. PMLR, 2021.
>
> [66] Han, Songqiao, et al. "Adbench: Anomaly detection benchmark." Advances in Neural Information Processing Systems 35 (2022): 32142-32159.
>
> [67] Cai, Jinyu, and Jicong Fan. "Perturbation learning based anomaly detection." Advances in Neural Information Processing Systems 35 (2022).

---

> > ### Comment · Reviewer_R9xH · 2023-08-21
> > **Response to authors**
> >
> > Thanks for the rebuttal. My questions have been well addressed.

---

### Official Review · Reviewer_3a9y · 2023-07-06

**Soundness:** 3 good
**Presentation:** 3 good
**Contribution:** 2 fair
**Rating:** 6
**Confidence:** 3

**Summary:**

- The authors proposed a selective predictor (learning to reject) for fully unsupervised setting in anomaly detection problems given an unsupervised anomaly detector.
- The proposed method is based on the theoretical supports and the threshold can be selected without any labeled data.
- The experimental results show that the proposed method can significantly reduce the cost of selective prediction.

**Strengths:**

- The proposed method is grounded on the theoretical supports that could be beneficial on generalization.
- The experimental sections are extensive and multiple ablation studies show its superiority across various settings.

**Weaknesses:**

- It seems like the proposed method is only applicable when the anomaly ratio is given. In some cases, anomaly ratio itself is not provided.
- It would be great if the authors can provide more extensive experiments when we have some labeled data in comparison to the baselines.

**Questions:**

1. EXCEED metric
- I can understand the brief concepts of the EXCEED metric.
- However, it would be good to further explain how the training data is perturbed.
- Also, it would be good to add additional explanations of Equation (1) - like the motivations of this equation.

2. Problem settings
- So, here, do we assume that we have an access to the contamination ratio (gamma)?
- If yes, can we extend this method without access to the contamination ratio?

3. With some labels
- As discussed in Related works, if we have some labels, we can easily optimize the rejection function using two ways that the authors explained.
- In that case, can we analyze how many samples do we need to have similar performance with the proposed unsupervised method?

**Limitations:**

Limitations are clearly stated.

---

> ### Author Rebuttal · Authors · 2023-08-08
>
> Dear Reviewer 3a9y,
>
> Thanks for the **very specific and helpful feedback**. We will address all your comments in the final version of the paper. Here is our response to your questions:
>
> 1.  ExCeeD uses a Bayesian formulation that simulates **bootstrapping** the training set as a form of perturbation. Equation 1 computes the confidence for a test score $s$ in two parts. **First**, it computes $\psi_n$ as the proportion of training scores lower than $s$. **Second**, it quantifies the probability that the model predicts anomaly for $s$ by estimating the proportion of times that $\psi_n > 1-\gamma$ (i.e., $s$ is in the top $\gamma$% of training scores) when simulating the bootstrapping of the training set.
>
> 2.  Yes, we assume that the **contamination ratio is given**, as stated in line 98. However, approaches exist to estimate it from a given dataset. This can even be done from a fully unlabeled dataset; see, for example, cite 50 in the paper.
>
> 3.  Experiment Q5 (lines 327-333) shows that our approach with a fully labeled training set would **only reduce the test cost by $0.6$%**, on average. We agree that analyzing on a theoretical level how many samples are needed to obtain a performance that is similar to the unsupervised one is interesting. This is certainly a good direction for **future work**, especially for unsupervised settings, because it would shed light on the number of labels needed to **justify the improvement**.

---

> > ### Comment · Reviewer_3a9y · 2023-08-18
> > **Thanks for the detailed response to my questions**
> >
> > Fully unsupervised settings (even without contamination ratio) would be the nice future work (or extensions).
> > Also, what I asked for Question (3) is more like an experimental analysis. Like how many samples do we need to collect instead of using the proposed method. If we just need less than 10 samples, some practitioners will just gather them instead of using this method.
> >
> > Anyway, I think this paper is an interesting paper and I will stand on my original score.

---

### Official Review · Reviewer_6n88 · 2023-07-20

**Soundness:** 3 good
**Presentation:** 3 good
**Contribution:** 3 good
**Rating:** 6
**Confidence:** 3

**Summary:**

This paper proposes an approach to perform learning to reject for anomaly detection in a completely unsupervised manner. The authors make three major contributions: (1) a novel theoretical analysis of a stability metric for anomaly detection, (2) a mechanism for designing an ambiguity rejection mechanism without any labeled data that offers strong guarantees, and (3) an evaluation of the proposed approach on an extensive set of unsupervised detectors and benchmark datasets. The authors show that their method outperforms several adapted baselines based on other unsupervised metrics and that their theoretical results hold in practice.

**Strengths:**

Originality: The paper proposes a novel approach to perform ambiguity rejection for anomaly detection in a completely unsupervised manner. The authors provide a novel theoretical analysis of a stability metric for anomaly detection and show that it has several previously unknown properties that are of great importance in the context of learning to reject.

Quality: The paper provides a thorough theoretical analysis of the proposed approach and demonstrates its effectiveness through experiments on an extensive set of unsupervised detectors and benchmark datasets. The authors also provide strong guarantees for their proposed method.

Clarity: The paper is well-written and easy to follow. The authors provide clear explanations of the proposed approach and the theoretical analysis.

Significance: The proposed approach addresses the challenge of uncertainty in traditional anomaly detectors and provides a solution through Learning to Reject. The authors show that their method outperforms several adapted baselines based on other unsupervised metrics and that their theoretical results hold in practice. The proposed approach has significant implications for anomaly detection in various domains.

**Weaknesses:**

The author may provide more intuition on how EXCEED works to estimate the stability.
The paper could benefit from a more detailed discussion of the limitations of the proposed approach and potential directions for future research.
While the paper provides a thorough theoretical analysis of the proposed approach, it could benefit from more detailed explanations of the experimental setup and results. Specifically, the paper could provide more information on the hyperparameters used in the experiments and how they were selected, as well as more detailed comparisons with other state-of-the-art methods.


**Questions:**

1. Is there any other method to estimate the detector's stability, in addition to EXCEED? If so, how does EXCEED outperform other methods?
2. In Definition 3.7, the cost function is defined as a simple addition. What if the cost function is of a more complex form?

**Limitations:**

The authors have adequately addressed the limitations and potential negative societal impact of their work.

---

> ### Author Rebuttal · Authors · 2023-08-08
>
> Dear Reviewer 6n88,
>
> Thanks for the **constructive feedback**. We will include a **more thorough overview** of how ExCeeD works in the final version of the paper to improve the readability of the paper. Lines 268 - 273 state that all the **hyperparameters** are set to their **default value** because we are operating in an unsupervised setting, meaning we **do not have labels** to tune them. Finally, here are our responses to your main concerns:
>
> 1.  [**Measuring Stability**] Apart from [47] which is included as a baseline in the paper, we are not aware of other methods that quantify a detector’s stability;
>
> 2.  [**Cost function**] Setting a proper cost function requires domain knowledge. In the learning to reject literature, most works use an additive cost function (see, e.g., the survey in [25]). Exploring other cost functions is a relevant area and if you are aware of other cost functions used in the literature, please let us know and we will include it in the final version of the paper.
>
> 3.  [**State-of-the-art rejection methods**] We would be happy to know what other methods the reviewer is referring to. To the best of our knowledge, there are no other algorithms for setting a rejection threshold without labels.

---

> > ### Comment · Reviewer_6n88 · 2023-08-16
> >
> > Thanks for the rebuttal! I have read it and will keep my recommendation.

---

### Decision · Program_Chairs · 2023-09-21

**Decision:**

Accept (poster)

**Comment:**

The authors consider the problem learning a rejector to discard anomalous samples *without* labeled data. The main idea is to apply a threshold to a confidence metric EXCEED proposed by prior work [48]. They offer supporting theory and showcase the efficacy of their approach in experiments.

Overall, the reviewers lean towards accepting the paper. I do however have a couple of important comments about the presentation, which the authors are strongly encouraged to address:
- The efficacy of the proposed method seems to rely on a *strong* modeling assumption that is **not stated** explicitly in the main paper (but mentioned in the prior work [48]). The authors assume a specific dependence between the base scoring function S and the label Y. More concretely, it is assumed that the label $Y|S=t$ follows a Bernoulli distribution with parameter $p_t = P(S \leq t)$. This assumption tying the label $Y$ to the scorer $S$ seems crucial to be able to construct a rejector *without labeled data*. It is important that this assumption is made very clear in the paper and its consequence in designing the rejector be discussed.
- The theoretical analysis appears to be a *finite-sample counterpart* to the asymptotic analysis already provided by the prior work [48]. Having made the above assumption tying the label $Y$ to the scorer $S$, the theoretical results seem to follow from application of *standard concentration inequalities*. Given that the theory is presented as a key contribution, it is important that the authors   highlight the technical difficulties in proving these results.

We urge the authors to address these comments in the camera-ready paper.